# Shapley Image Explanations With Data-Aware Binary Partition Trees

## Abstract

Extracting a visual interpretation of a learned representation of a machine learning model applied to image data is a relevant task in eXplainable AI (XAI). Effective visual explanations must reveal how specific features within the learned representation contribute to the model's predictions. Pixel-level feature attributions are a valuable tool for this, as they highlight the regions in the image that are most influential in the classification process. The hierarchical Owen approximation of the Shapley values has proved to be an effective strategy for this task. However, existing approaches lack data-awareness, leading to poor alignment between the pixel-level attributions and the actual morphological features of the classified image. This paper introduces *ShapBPT*, a novel XAI method that computes the Owen approximation of the Shapley coefficients following a *data-aware* binary hierarchical coalition structure, generated from the Binary Partition Tree computer vision algorithm. By aligning with the morphological features of the image, the proposed method significantly enhances the identification of relevant image regions. Experimental results confirm the effectiveness of the proposed method.

## 1 Introduction

In the field of AI, understanding how a black-box machine learning (ML) model classifies images is a task of critical importance to extract the representations the model has learned from the data. We consider the problem of attributing importance scores to individual pixels in an image, which reflect their contribution to the model's decision-making process. This task is commonly referred to as *explaining* a black-box machine learning (ML) model classifying images.

In recent years, several notable practical approaches were developed to address this task. A pioneering approach to this task was LIME (Local Interpretable Model-agnostic Explanations), which reformulates the problem of explaining image classifications by leveraging an image segmentation algorithm. This transformation passes from pixel-level attribution values to segment-level scores, computed using a simple linear regression model (Ribeiro et al., 2016). Although lacking theoretical guarantees, the effectiveness of LIME lies in its ability to potentially pre-identify relevant regions through segmentation.

Another influential method is SHAP (SHapley Additive exPlanations), which applies game-theoretic principles to ML explainability. SHAP combines a feature removal (masking) technique (Lundberg & Lee, 2017) together with the use of a simple hierarchical image partitioning (Lundberg, 2020). Providing explanations over hierarchical image structures leverages multi-scale image features, which provides better approximations of the representations learnt by the classification model.

In general, it is reasonable to assume that in any image classification task, an effective ML model needs to learn some form of structured representation that combines some arbitrarily complex but distinct morphological characteristics of the classified objects (shape, texture, color continuity, etc), as we assume that the model *has* learned to recognize structured patterns from the image data. Consequently, adopting hierarchical partitions that are adaptive and data-aware can improve the model's interpretability by aligning more closely with the learned representations, as long as the partitions are flexible and adaptive and not imposed a-priori (as we cannot assume *which* structured representation the model has learnt). Such an approach ensures that the explanations reflect the underlying features in a way that is both accurate and interpretable, without distorting the model's internal hierarchy of representations.

This paper provides the following contributions:

1. A novel hierarchical model-agnostic eXplainable AI (XAI) strategy that integrates an adaptive multi-scale partitioning algorithm with the Owen approximation of the Shapley coefficients. We identify in the BPT (Binary Partition Tree) algorithm of Salembier & Garrido (2000) a highly valuable candidate for such task. This approach overcomes the limitations of the inflexible hierarchies adopted by existing state-of-the-art methods like SHAP.

2. An empirical assessment of the proposed method showcasing its efficacy across various scoring targets, in comparison to established state-of-the-art XAI methods.

We show that the proposed approach surpasses existing Shapley-based model-agnostic XAI methods that do not leverage on data-awareness, and at the same time it achieves a significantly faster convergence rate. This efficiency stems from the fact that, on average, fewer recursive applications of the Owen formula (i.e. expansions of the partition hierarchy) are needed to accurately localize objects when using a *data-aware* partition hierarchy, such as the proposed BPT hierarchy, compared to other hierarchies. As far as we know, this is the first XAI method that combines the Owen formula with a data-aware partition hierarchy for image data, and with this paper we prove the effectiveness of this combined strategy for interpreting ML classifiers.

## 2 METHODOLOGY

A fundamental ML objective is to discover a function, denoted as $f : \mathcal{X} \rightarrow \mathcal{Y}$, that effectively approximates a response $y \in \mathcal{Y}$ corresponding to a given input $x \in \mathcal{X}$. For the sake of simplicity, we will assume $\mathcal{Y} \subseteq \mathbb{R}$ and $\mathcal{X} \subseteq \mathbb{R}^n$. In many practical cases only a subset of $x$ significantly influences the resulting response $y = f(x)$. Understanding the relative importance, or *contribution*, of each component $x_i$ of $x$ in determining the value of $y$ by $f$ is a central problem in XAI. One important approach proposed by Covert et al. (2021) for assessing these contributions is through a technique known as *feature removal* or *masking*, wherein certain values of $x$ are replaced with values from a specified context-dependent background set. Let $\nu_{f,x} : 2^{|\mathcal{X}|} \rightarrow \mathcal{Y}$ be a *masking function* for $f(x)$, where $\nu_{f,x}(S)$ represents the resulting model evaluation when only the elements in the subset $S$ of $x$ are retained, while the remainders are masked. Hereafter we will denote $\nu_{f,x}$ as $\nu$.

### 2.1 SHAPLEY VALUES FOR HIERARCHICAL COALITION STRUCTURES (HCS)

We consider the setup of a $n$-coalition game $(\mathcal{N}, \nu)$, which can be considered analogous to an importance scores attribution task in XAI (Rozemberczki et al., 2022). The finite set $\mathcal{N} = \{1, \ldots, n\}$ is the set of players (*features*). Each non-empty subset $S \subseteq \mathcal{N}$ is a *coalition*, and $\mathcal{N}$ is itself the *grand coalition*. A *characteristic function* $\nu : 2^n \rightarrow \mathbb{R}$ assigns to each coalition $S$ a (real) *worth value* $\nu(S)$, and it is assumed that $\nu(\varnothing) = 0$[1]. A *marginal contribution* of a player $i$ to a coalition $S$ (assuming $i \notin S$) is given by

$$\Delta_i(S) = \nu(S \cup \{i\}) - \nu(S) \tag{1}$$

Semivalues (Dubey et al., 1981) are weighted sums of marginal contributions (1), and they were proposed to address the issue of fairly distributing the total worth $\nu(\mathcal{N})$ of the grand coalition $\mathcal{N}$ among its members. The Shapley value, a well-known semivalue introduced in Shapley (1953), demonstrates favorable axiomatic properties and it has been used effectively to explain ML models (Rozemberczki et al., 2022).

A fixed a-priori *coalition structure* (López & Saboya, 2009; Owen, 2013; 1977) for the $\mathcal{N}$ players is a finite set $\{T_1 \ldots T_m\}$ of $m$ partitions of $\mathcal{N}$ (i.e. $\cup_{k=1}^m T_k = \mathcal{N}$, and $T_i \cap T_j \neq \varnothing \Leftrightarrow i = j$). Elements $T_i$ are usually called *partitions*, *coalitions*, *teams* or *unions*.

We consider a recursive definition of a hierarchical coalition structure, where each partition $T$ can be either an *indivisible partition* or a *sub-coalition structure* itself $T = T_1 \cup \ldots \cup T_m$. Let $T\downarrow$ be the (downward) recursive partitioning of $T$, defined as

$$T\downarrow = \begin{cases} \{T_1 \ldots T_m\} & \text{if } T \text{ admits a sub-coalition structure} \\ \bot & \text{if } T \text{ is indivisible} \end{cases} \tag{2}$$

---

[1]By translating the equation system, it is always possible to ensure $\nu(\varnothing) = 0$.

We denote with $\mathcal{T}$ the HCS root, and assume w.l.o.g. that $\mathcal{T}$ contains all the elements of $\mathcal{N}$.

A special case of HCS happens when each sub-coalition structure is made by two partitions, i.e. the hierarchy forms a binary tree. We refer to these structures as *binary hierarchical coalition structures* (BHCS). In that case the recursive downward partitioning of $T$ can be simplified as

$$T\downarrow = \begin{cases} \{T_1, T_2\} & \text{if } T \text{ admits a binary sub-coalition structure} \\ \bot & \text{if } T \text{ is indivisible} \end{cases} \tag{3}$$

## 2.2 THE OWEN APPROXIMATION OF SHAPLEY VALUES FOR BINARY HCS

Computing Shapley values has exponential time complexity, which is unfeasible for image data with hundreds or thousand of features (pixels). An approximate approach, introduced by Owen (1977) can be used to drastically reduce the complexity by grouping features into hierarchical coalitions. This concept has been pioneered for image data by the SHAP Partition Explainer (Lundberg, 2020; Shrikumar et al., 2017; Lundberg & Lee, 2017).

A *coalition value* $\Omega_i(\mathcal{T})$ represents the worth of player $i$ in a game with coalition structure $\mathcal{T}$, and is known as the Owen coalition value (Owen, 1977). Computing coalition values over a binary HCS $T$ as defined in (3) can be done with a recursive formula

$$\Omega_i^{\text{B}}(Q, T) = \begin{cases} \frac{1}{2}\Omega_i^{\text{B}}(Q \cup T_2, T_1\downarrow) + \frac{1}{2}\Omega_i^{\text{B}}(Q, T_1\downarrow) & \text{if } T\downarrow = \{T_1, T_2\} \\ \frac{1}{|T|}\Delta_T(Q) & \text{if } T \text{ is indivisible} \end{cases} \tag{4}$$

s.t. $\Omega_i(\mathcal{T}) = \Omega_i^{\text{B}}(\varnothing, \mathcal{T})$. The former case of Eq. (4) deals with coalitions $T$ that admit a sub-coalition structure $T\downarrow \neq \bot$. We assume, for notational simplicity and without loss of generality, that $i \in T_1$. The latter case of Eq. (4) deals with indivisible coalitions. In that case, the formula assigns a single coalition value to all players inside the coalition $T$, divided uniformly among all the members of $T$.

In the rest of the paper, we will refer to the Owen approximation of the Shapley values simply as Shapley values. Note that Eq. (4) is not found in published literature (as far as we know), and its complete derivation is therefore provided in Appendix A.1.

**Theorem 1.** ***Computational cost.*** *Consider a BHCS consisting of a balanced tree of depth $d$. The time complexity of Eq. (4) is in the order of $O(4^d)$ evaluations of the $\nu$ function.*

*Proof.* In Appendix A.2. $\qquad\qquad\square$

Theorem 1 highlights the exponential cost of Eq. (4). However, practical implementation of Eq. (4) do not rely on expanding a fully balanced BHCS tree to a fixed depth $d$. Instead , they employ an adaptive splitting strategy that is not limited to balanced trees. In this adaptive case, a total budget $b$ of evaluations of the masked model $\nu$ is allocated. The adaptive algorithm then iteratively explores the tree hierarchy, at each iteration splitting the partition $T$ that maximizes the sum of its Shapley values, $\sum_{i \in T} \Omega_i^{\text{B}}(\varnothing, \mathcal{T})$. Each partition split requires 2 model evaluations. A pseudo-code of this adaptive algorithm is provided in Appendix A.3. Despite adaptively ignoring certain coalitions, the cost of exploring the hierarchy at depth $d$ remains exponential, as stated in Theorem 1.

## 3 HIERARCHICAL COALITION STRUCTURES FOR IMAGE DATA

Calculating Owen coalition values for image data necessitates a well-defined hierarchical structure that captures both spatial relationships and image semantics. Our approach is aimed at addressing limitations in existing methods, by emphasizing the importance of these factors in coalition formation. We therefore consider and compare both *data-agnostic* and *data-aware* approaches.

In a *data-agnostic* approach, partitions are created based on simple geometric divisions, like grids or quadrants. The *Axis Aligned hierarchy* (AA hereafter) is one such approach to building hierarchical coalition structures, adopted by the SHAP's Partition Explainer (Lundberg, 2020) and by h-SHAP (Teneggi et al., 2022) In an AA hierarchy, each partition $T$ corresponds to a rectangular region within the image, and $T\downarrow$ splits the rectangular region of $T$ in half along the longest axis. This splitting process continues until individual, indivisible regions (unitary regions, with a single pixel)

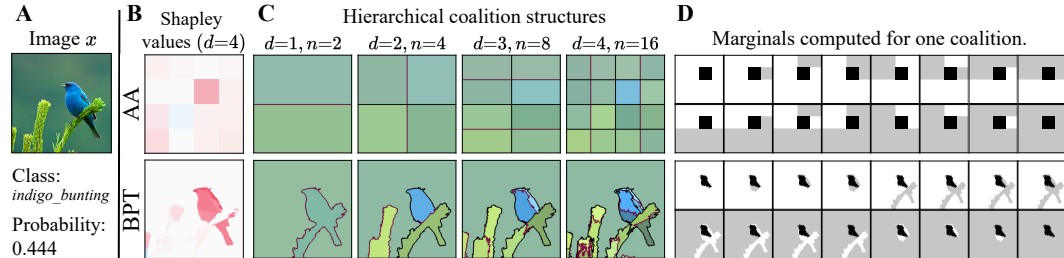

Figure 1: AA and BPT coalition structures for a sample image, explanations from a ResNet50 model.

are reached. The main limitation of this approach is that properly localizing the relevant regions within an image may require a large number of recursive evaluation of the Owen's formula (4), and this evaluation follows the $O(4^d)$ time cost of Theorem 1.

In a *data-aware* approach, morphological features within the image guide the partitioning process. This approach, pioneered by Ribeiro et al. (2016) with LIME, utilizes a pre-defined segmentation algorithm to divide the image into regions (patches). Although effective, the main limitation is the lack of an effective feedback loop within the explanation method. If the segmentation is inaccurate, the resulting explanation is poor, and there is no opportunity for refinement.

A notable algorithm for hierarchical segmentation, that fits well with Eq. (4), is the *Binary Partition Tree* (BPT) (Randrianasoa et al., 2018), originally developed for multiscale image representation in MPEG-7 encoding (Salembier & Garrido, 2000). The intuitive principle is that portions of an image with similar color and coherent shape are highly likely to have similar Shapley values, thereby maximizing the effectiveness of Eq. (4).

Theorem 1 shows that the Owen approximation cost increases rapidly if a large number of coalitions need to be evaluated recursively. Therefore, an effective BHCS needs to satisfy these requirements:

R1 As few recursive cuts as possible to reach the relevant regions, as each cut increases the required evaluation budget $b$ exponentially;

R2 Partitions should not be fixed, since the relevant regions are not known in advance.

AA hierarchies do no satisfy R1, and most a-priori segmentation algorithms do no satisfy R2. The solution that we propose, which constitutes the main contribution of this paper, is a novel hybrid method that finally statisfies the two aforementioned requirements by combining a dynamic a-priori hierarchical coalition structure (the BPT) aligned with the morphological features of the image (e.g., color uniformity, pixel locality) together with an a-posteriori splitting strategy based on the distribution of Shapley values (as in the Partition Explainer). This combination results in fewer recursive applications of the Owen formula needed to accurately localize objects, compared to data-agnostic coalition structures. As we shall see in the experimental section, this approach gets a significantly faster convergence than other Shapley-based methods, paired with accurate shape recognition of the classified objects.

### 3.1 GENERATING BPT HIERARCHIES.

A *BPT hierarchy* captures how we can progressively merge (Randrianasoa et al., 2018) the $n$ pixels of an image $x$ into larger regions, forming a quasi-balanced binary tree. Such tree is built bottom-up, starting from an initial coalition structure $\mathcal{T}_{[1]} = \{T_1 = \{1\} \ldots T_n = \{n\}\}$ made by $n$ unitary and indivisible partitions, where the features $1 \ldots n$ represents the individual pixels of the image. Two partitions $T_i, T_j \in \mathcal{T}_{[k]}$ are *adjacent* if there is at least one pixel of $T_i$ that is adjacent to a pixel of $T_j$ in the image. The BPT construction involves merging adjacent partitions iteratively. A *coalition merge* of $\mathcal{T}_{[k]}$ is a new coalition structure $\mathcal{T}_{[k+1]}$ where two adjacent partitions $T_i, T_j \in \mathcal{T}_{[k]}$ are removed and replaced by a new partition $T_{n+k}$, s.t. $T_{n+k} = T_i \cup T_j$ and $T_{n+k}\downarrow = \{T_i, T_j\}$.

The two adjacent partitions $T_i, T_j$ of $\mathcal{T}_{[k]}$ being merged are selected by minimizing a *data-aware* distance functions. While multi-criteria BPT are possible (Randrianasoa et al., 2021), we focus on a

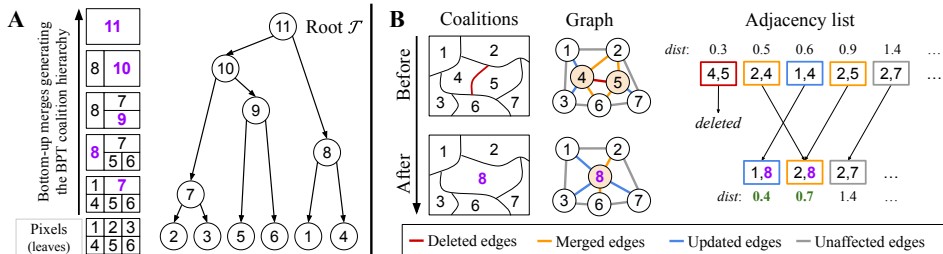

Figure 2: **(A)** BPT generating by bottom-up merging coalitions from the pixels (1–6) to the to the root (11). **(B)** Details of one merging step $T_8 \downarrow = \{T_4, T_5\}$ on some arbitrary coalition structure.

simple distance based on the intuitions found in Randrianasoa et al. (2018) and defined as

$$dist(T_i, T_j) \; = \; dist^2_{\text{color}}(T_i, T_j) \cdot area(T_i, T_j) \cdot \sqrt{perim(T_i, T_j)} \qquad (5)$$

where $dist^2_{\text{color}}(T_i, T_j)$ is the sum of the squared color ranges of $T_i \cup T_j$, for all color channels, and $area(T_i, T_j)$ and $perim(T_i, T_j)$ are the area and the perimeter of $T_i \cup T_j$, respectively. This function is a heuristic criterion that balances together color similarity and shape regularity (perimeter). The area improves the construction of a (semi)-balanced tree, which is a desirable feature of such trees.

A *merging sequence* $\mathcal{T}_{[1]} \rightarrow \mathcal{T}_{[2]} \rightarrow \ldots \rightarrow \mathcal{T}_{[n]}$ is a sequence of $n-1$ coalition merges. The sequence ends with the coalition structure $\mathcal{T}_{[n]} = \{T_{2n-1}\}$, having a single partition with all pixels. At this point, all non-unitary partitions $T$ at any point in the merging sequence admit a binary sub-coalition structure $T\downarrow$. Therefore, the BPT $\mathcal{T}_{[n]}$ satisfies Eq. 3, and may become the root $\mathcal{T}$ of the BHCS. An illustration of the algorithm generating the BPT merging sequence is shown in Fig. 2/A, where the unitary partitions are merged, one by one, until all pixels are merged into the root $\mathcal{T}$. The operations needed to perform a single merging step are illustrated in Fig. 2/B, while a detailed pseudo-code of the BPT algorithm is provided in Appendix A.4.

*Example* 1. *Figure 1 shows a sample image (A) alongside its Shapley explanations (B) obtained from applying Eq. (4) on the AA and BPT hierarchical coalition structures (C), up to a predetermined depth value $d = 4$. The first four depth levels of the tree hierarchy are depicted in (C), to show how the BPT partitions are data-aware. In these explanations, each hierarchical coalition value is computed through weighted sums of the eight marginals $\widehat{\varphi}_i(Q, T)$, and those eight marginals for the highest value are depicted in (D), where $Q$ and $T$ represent the grey and black regions, respectively. Coalitions depicted in (D) are obtained by the application of Eq. (4).*

## 4 EXPERIMENTAL ASSESSMENT

We present a comparative analysis of the performance of the proposed Shapley method using BPT partitions, alongside other state-of-the-art image explainers.

**Comparison scores.** We consider a quantitative evaluation of the methods using six different scores, summarized hereafter. The first two are the established metrics from Petsiuk et al. (2018). These two *area-under-curve* scores measure how well the explanation coefficients (represented by Shapley values) in rank order align with the black-box model's output. Let $S^{[q]} \subseteq \mathcal{N}$ be the subset of the first $q$-th quantile of elements from $\mathcal{N}$ with the largest Shapley values. Define

$$AUC^+ = \int_0^1 \nu\big(S^{[q]}\big)\,\mathrm{d}q, \qquad AUC^- = \int_0^1 \nu\big(\mathcal{N} \setminus S^{[q]}\big)\,\mathrm{d}q \qquad (6)$$

With this definition $AUC^+$ (and $AUC^-$) evaluate the model's behavior as features are progressively included ($AUC^+$) or excluded ($AUC^-$) from an empty set (for inclusion) or the full set (for exclusion). Intuitively, both scores assess if features with higher Shapley values are indeed more important for the model's prediction.

We extend the previous scores by quantifying how fairly the sum of the Shapley values for the features $S$ contribute to the model output $\nu(S)$. Let $\eta(S)$ be the sum of Shapley values for any

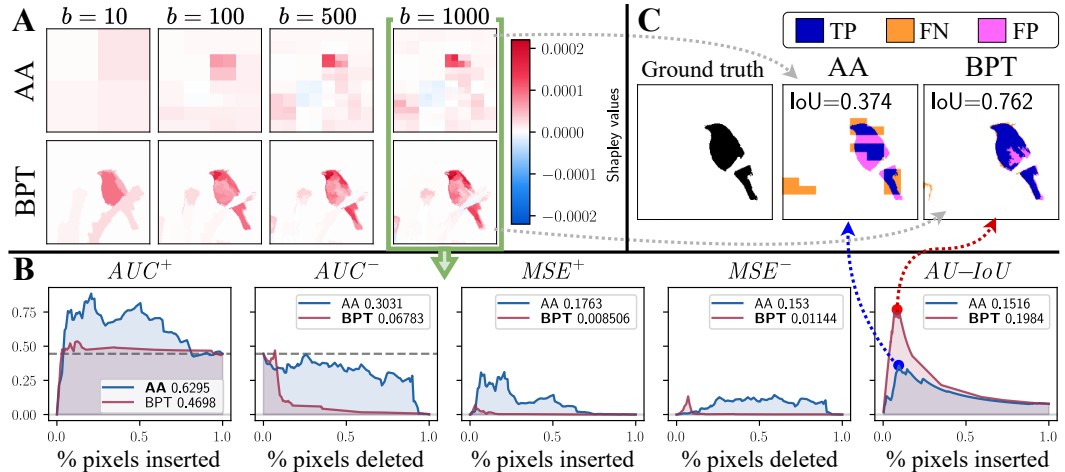

Figure 3: Shapley values for AA and BPT coalition structures, for different values of the budget $b$.

subset $S \subseteq \mathcal{N}$. Ideally, the change in model output $\nu(S)$ should be directly proportional to the sum of Shapley values of the included features $\eta(S)$, reflecting the *efficiency* axiom (Rozemberczki et al., 2022). Therefore, we can consider the difference $\nu(S) - \nu(\varnothing)$ as an error, and take its squared mean. The scores $MSE^+$ and $MSE^-$ follow the same insertion/deletion logic of Eq. (6) while also quantifying *how proportionally* the assigned Shapley values translate into their actual influence on the model's output.

$$MSE^+ = \int_0^1 \left( \nu(S^{[q]}) - \eta(S^{[q]}) \right)^2 \mathrm{d}q, \quad MSE^- = \int_0^1 \left( \nu(\mathcal{N} \setminus S^{[q]}) - \eta(\mathcal{N} \setminus S^{[q]}) \right)^2 \mathrm{d}q \quad (7)$$

We consider also two metrics that are specific for the *Visual Recognition Challenge* (VRC) problem. Assume that there is a subset $G \subseteq \mathcal{N}$ that defines the features that should ideally contribute to the classification, i.e. $\nu(G) = \nu(\mathcal{N})$. Assume that $G$, the *ground truth*, is known for the evaluation. An explanation is a *perfect recognition* if there is a threshold $q$ for which $S^{[q]} = G$. Consider the standard *Intersection-over-Union* score $J(A, B) = \frac{|A \cap B|}{|A \cup B|}$ and define

$$AU\text{-}IoU = \int_0^1 J(S^{[q]}, G) \, \mathrm{d}q, \qquad max\text{-}IoU = \max_{q \in [0,1]} \left( J(S^{[q]}, G) \right) \quad (8)$$

The Area Under IoU curve ($AU\text{-}IoU$) score (Gangopadhyay et al., 2023) is the area of the curve of the IoU value for all the thresholds $q \in [0, 1]$, while $max\text{-}IoU$ is the curve maximum. The $AU\text{-}IoU$ is maximal if the explanation is a perfect recognition, and in such case $max\text{-}IoU = 1$.

*Example 2. Figure 3 shows the Shapley values computed using Eq. (4) on the AA and BPT coalition structures, by refining the most significant coalition using a budget b of model evaluations (A), for four budget values of 10, 100, 500 and 1000 samples, respectively. The five plots (B) depict the AU curves for the five considered AUC scores (6), (7) and (8), for the case b=1000. The area identified by the threshold q obtaining the maximal IoU is depicted in (C). In the example, BPT demonstrates significantly faster convergence and improved object region recognition w.r.t. AA.*

**Compared methods.** We run a comparative analysis using several state-of-the-art XAI methods, categorized into two groups. The first group comprises Shapley-based methods, chosen for their compatibility with our proposed approach. They include: **BPT**-$b$: our proposed Shapley explanation method with BPT partitions, with sample budgets $b$ of 100, 500, and 1000 samples; **AA**-$b$: the SHAP Partition Explainer (Lundberg, 2020), utilizing Axis-Aligned partitions with $b$ of 100, 500, and 1000 samples; **LIME**-$k$: LIME[2] explanation (Ribeiro et al., 2016) with $k$ segments (with $k$ being 50, 100 and 200) and a budget $b = 5 \cdot k$.

---

[2]Although LIME does not generate Shapley values, it has theoretical and practical similarities to them (Lundberg & Lee, 2017).

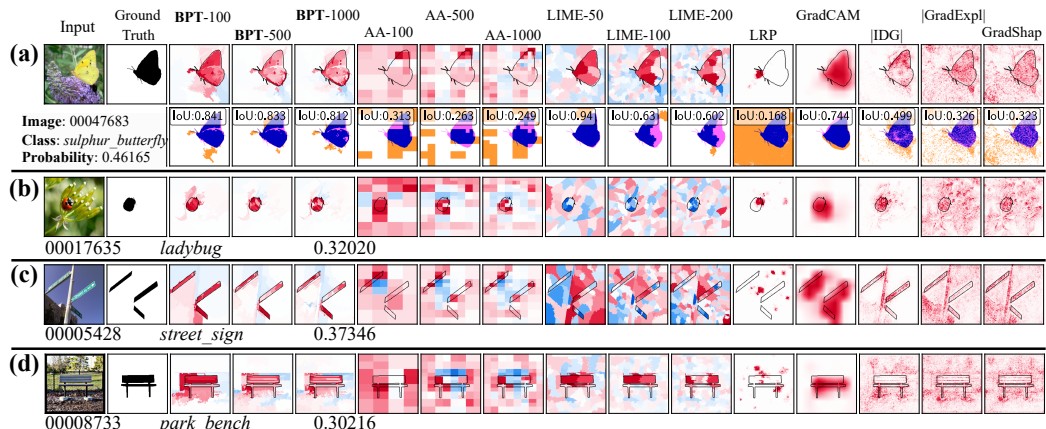

Figure 4: Saliency maps for a few ImageNet-S$_{50}$ images, classified by the ResNet50 model.

The second group consists of gradient-based methods, included in our analysis due to their widespread usage. They include: **GradExpl**: The Gradient Explainer from the SHAP package (Lundberg & Lee, 2017), using the default of 20 samples; **GradCAM**: The Gradient-weighted Class Activation Mapping introduced by Gildenblat & contributors (2021); **IDG**: The Integrated Decision Gradient method proposed by Walker et al. (2024); **LRP**: Layer-wise relevance propagation of Bach et al. (2015); Ancona et al. (2018) from the Captum library; **GradShap**: Gradient Shap of Sundararajan et al. (2017).

Explanations from LIME and gradient-based methods are normalized to the $\nu(\mathcal{N}) - \nu(\varnothing)$ value before computing the $MSE$ scores. For *GradExpl* and *IDG*, we utilize the absolute values of the produced explanations, resulting in superior scores compared to the signed values.

| Name | Dataset | Model | Short description | Reference |
|------|---------|-------|------------------|-----------|
| E1 | ImageNet-S$_{50}$ | ResNet50 | Common ImageNet setup | Fig. 5 |
| E2 | ImageNet-S$_{50}$ | Ideal | Controlled setup for exact IoU | Appendix A.7 |
| E3 | ImageNet-S$_{50}$ | ResNet50 | Multiple replacement values | Appendix A.8 |
| E4 | ImageNet-S$_{50}$ | VGG-16 | Common ImageNet setup | Appendix A.9 |
| E5 | ImageNet-S$_{50}$ | Swin-ViT | Vision Transformer model | Appendix A.10 |
| E6 | MVTec | VAE-GAN | Explainable Anomaly Detection | Appendix A.11 |
| E7 | CelebA | CNN | Facial attributes localization | Appendix A.12 |

Table 1: Summary of the experiments

A summary of the experiments included in this paper is provided in Table 1. We focus on the experiment E1, which is a typical ImageNet setup that is commonly used to benchmark explainable AI methods. The remaining experiments are reported in the Appendix.

**Experiment E1.** We use the *1K-V2* pretrained (Vryniotis, 2021) ResNet50 (He et al., 2016) model found in the PyTorch library, with accuracy $80.858\%$. Masking is performed by replacing pixels with uniform gray. We consider the ImageNet-S$_{50}$ dataset of Gao et al. (2022), which features precise ground truth masks for a few selected images. For simplicity, we consider the images for which the ground truth is available for the top predicted class, resulting in 574 images.

**Saliency maps.** Looking directly at the saliency maps of the explanations generated by the tested models allows us to get a first intuition of the characteristics of the BPT method. Figure 4 shows a few selected examples. Each row reports the image, the ground truth $G$, and the saliency maps for the fourteen tested methods in the **E1** setup. The boundaries of $G$ are drawn overlapped to every saliency map, to help identify the object. To illustrate the evaluation process, for the first image, we also report the optimal IoU w.r.t. $G$. In general BPT explanations (columns 3–5) show a better tendency of identifying the partition borders, cutting the recognized object from the background. In that sense, they share similarities with the explanations of LIME, but without the typical LIME

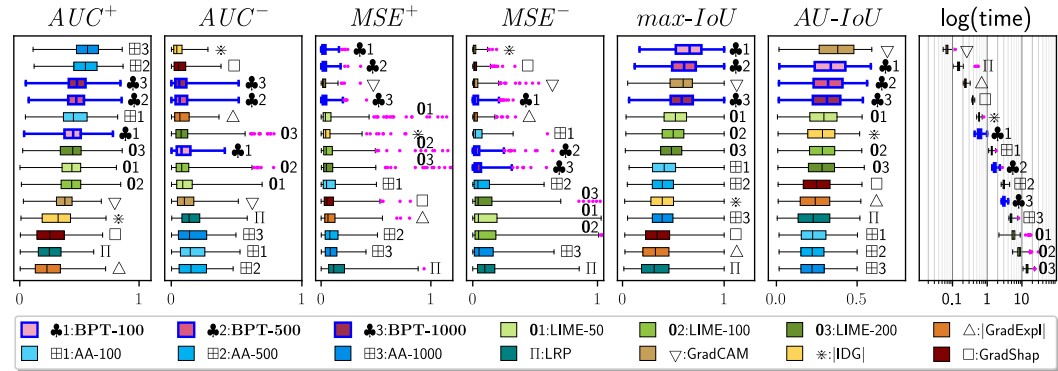

Figure 5: Results for the six metrics across 574 images from the ImageNet-$S_{50}$ dataset, with methods sorted to display the highest-performing one atop each column, for the experiments E1. Higher is Better for $AUC^+$, and $max - IoU$ and Lower is Better for the remaining ones.

noise, and without relying on a fixed, inflexible segmentation. Moreover BPT explanations look a lot more in accordance to those of GradCAM, but without the blurriness that the latter adds. While all the tested methods seem to somewhat agree on the recognition area, the practical behaviour of BPT seems in line with its theoretical assumption that splitting the image partitions following the morphological image boundaries leads to better object recognition, and better separation from the background. Additional saliency maps for **E1** are included in Appendix A.6.

**Numerical results.** Figure 5 reports the results for **E1**, with one table for each of the six metrics, plus one for the evaluation time[3] (logscale). Scores are drawn as boxplots (treating values outside 10 times the interquantile range as outliers, drawn as fuchsia dots), with a method symbol on the right (see the legend for the mapping). We conducted one-way ANOVA tests for each score to assess whether the null hypothesis ($H_0$) of equal means across all sample populations could be rejected, with a p-value threshold of $0.05$. All scores reject $H_0$, implying that there is sufficient data to suggest that the result is significant.

In **E1**, BPT is positioned close or at the top of every score. In this case, AA has a slightly better $AUC^+$ score, but a worse $AUC^-$ score than BPT. Interestingly, GradCAM and IDG get very low $MSE$ errors, which is unexpected since these are not Shapley-based methods and do not obey any efficiency axiom. The BPT method seems to be particularly effective at the IoU scores *max-IoU* and *AU-IoU*, which can be explained by the capacity of recognizing the borders of the objects, by following a data-aware hierarchy. Only GradCAM reaches similar IoU scores, but in practice the localization of GradCAM is more blurred and fuzzy (this limitation is apparently not well captured by the two IoU scores).

As a side note, observe that the IoU scores make the assumption that the ground truth $G$ is actually aligned with the learned representation of the model. This is likely to be an approximation, as it is known that deep learning models on problems like ImageNet tend to learn weak correlations between objects, and focus on details. However, this approximation affects uniformly all XAI methods, as they all explain the same model, and in principle should not introduce a bias favoring some specific XAI method. We designed a separate experiment **E2**, reported in Appendix A.7, that avoids this issue by using a linear model perfectly aligned with the ground truth. The $AUC$ and $MSE$ scores are unaffected by this approximation, as they do not rely on any ground truth.

Descriptions and results of the remaining experiments in Table 1 are presented and discussed in the appendix (A.7–A.12). While these findings may not be deemed conclusive, we observe that BPT outperforms AA in the region localization problem and in several metrics, while also achieving effective explanations with very little budget – sometimes even an order of magnitude less.

---

[3]All reported times were computed with an Intel Core9 CPU, an Nvidia 4070 GPU, and 16GB of RAM.

## 5 DISCUSSION ON LIMITATIONS AND OTHER RELATED WORKS

The proposed method, as repeatedly mentioned in the text, combines the SHAP Partition Explainer of Lundberg (2020) with the partition hierarchy of the BPT algorithm of Salembier & Garrido (2000). The rationale behind this combination lies in the notion that image regions sharing similar morphological characteristics are likely to exhibit comparable Shapley contributions. If this rationale is not satisfied for a partition, such partition can be further subdivided, progressively localizing the regions activating model recognition. The proposed approach is general, but it is particularly useful for the Visual Recognition Challenge (Russakovsky et al., 2015) problem. However, the effectiveness of this approach is based on the assumption of a correlation between image morphological features and their corresponding explanations. While the assumption may appear reasonable, assessing its complete impact is challenging.

A close concern is related to the introduction of potential biases. After extensive experimentation with the proposed method, we hypothesize that BPT partitions do not introduce significant biases w.r.t. other Owen approximations. However, we have not formally quantified this assertion, leaving it as a subject for future research.

We considered also the *h-Shap* approach of Teneggi et al. (2022), which exhibits faster convergence than the one derived in Theorem 1. Unfortunately, the different definition of the *object recognition task* makes the comparison challenging, and we have not included it in the evaluation. However, we believe that also *h-Shap* would greatly benefit from using BPT partitions.

We used the *quickshift* algorithm to generate the fixed a priori partitions for LIME. We also evaluated the more recent *SegmentAnything* partitioning algorithm, which offers some improvements over *quickshift*, albeit at the cost of being significantly slower. However, the rigidity of working with a priori partitions that may not align with the model's internal representation persists, a limitation that is addressed by the proposed BPT approach.

We initially considered incorporating the *relevance mass and rank accuracy* scores from Arras et al. (2022) into our analysis of the experiment results. However, we ultimately decided against it, as these metrics rely on non-negative values, which are incompatible with Shapley values.

While Eq. (5) provides reasonable partitionings in the experimental setup, it is also well recognised to be a critical (Randrianasoa et al., 2021) component of the BPT algorithm. A complete analysis and optimization of this heuristic equation has not beed carried out, and it is left for a future work.

As a side note, we empirically observed that the $AUC^+$ and $AUC^-$ scores (Petsiuk et al., 2018), which are considered among the gold standards for XAI evaluation (Nauta et al., 2023), do not always align with our intuition. For example, in Figure 3/B, $AUC^+$ shows a significant overshoot above the $\nu(\mathcal{N})$ prediction value. While this is beyond the goal of this paper, we believe further investigation into this class of XAI scores is needed, particularly regarding the behavior of overshooting beyond the prediction range of $\nu(\mathcal{N})$ and $\nu(\varnothing)$ (more details in Appendix A.8).

## 6 CONCLUSIONS

This paper introduces a novel eXplainable AI method, named ShapBPT, that generates image explanations by computing the the Owen approximation of the Shapley coefficients following a data-aware Binary Partition Tree hierarchy. We provide the formulation of the method, including the approximation at the indivisible partitions, its computational cost, and the algorithms. An evaluation is performed on multiple settings, models and datasets, with a full scale comparison with other state-of-the-art XAI methods. We believe that our method produces explanations that are noticeably better both visually and quantitatively compared to existing methods, as they are built following a coalition structure that is hierarchically and adaptively expanded to better follow the morphological features of the image data, which are assumed to be the representation learnt by the model.

**Reproducibility Statement:** We provide as Supplementary Material: I) the ShapBPT library code; II) all the notebooks needed to reproduce the benchmark and generate the figures included in this paper; III) instructions on how to obtain the datasets and how to install and run the whole benchmark. In addition we provide a link to an anonymous repository containing all the trained model weights and all the precomputed results. All supplementary material will be made publicly available upon acceptance.

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

# A APPENDIX

## A.1 DERIVATION OF EQUATION (4)

We present a clear formulation of the Owen approximation of Shapley values within a hierarchical coalition structure, as this specific approach appears to be absent from existing published literature. To ease our formulation, we start from a simple extension of the Shapley formula:

$$\varphi_i(Q, \mathcal{N}) = \sum_{S \subseteq \mathcal{N} \setminus \{i\}} \frac{1}{n \cdot \binom{n-1}{|S|}} \Delta_i(Q \cup S) \tag{9}$$

where $n$ is the cardinality of $\mathcal{N}$. Eq. (9) assigns a unique distribution of the total worth $\nu(\mathcal{N})$ generated by cooperation among players in a coalition game, and is extended by assuming that all coalitions $S$ are supported by a persistent set of players $Q$. The regular Shapley value (Shapley, 1953, Eq.12) are obtained from (9) as $\varphi_i(\varnothing, \mathcal{N})$. The persistent set $Q$ is used for the Owen approximation.

The Owen coalition value (Owen, 1977) is an extension of the Shapley value, and it is a quantity $\Omega_i(\mathcal{T})$ that represents the worth of player $i$ in a game with coalition structure $\mathcal{T}$. The original formulation for a two-level coalition structure hierarchy[4] works as follows. Consider a player $i$ belonging to team $T_j \in \mathcal{T} \downarrow$. Then

$$\Omega_i(\mathcal{T}) = \sum_{\substack{H \subset M \\ j \notin H}} \sum_{\substack{S \subset T_j \\ i \notin S}} \frac{h!(m-h-1)! \, s! \, (t_j - s - 1)!}{m! \, t_j!} \Delta_i(Q_H \cup S) \tag{10}$$

where $M = \{1 \ldots m\}$ is the set of structured coalition indices of $\mathcal{T}$, $Q_H = \bigcup_{k \in H} T_k$, and the values $h$, $s$, $t_j$ are the cardinalities of the sets $H$, $S$ and $T_j$, respectively.

Eq. (10) can be seen as a two-level Shapley value, where inside a team $T_j$ all coalitions are possible, but once a coalition $S \subset T_j$ is formed, only a restricted *all-or-nothing* form of cooperation with the other teams is possible. In fact, it is possible to rewrite (10) by explicitly identifying the Shapley value for the subsets $S$ of $T_j$. By doing so with (9) and applying simple algebraic transformations, we get

$$\Omega_i(\mathcal{T}) = \sum_{H \subseteq M \setminus \{j\}} \frac{1}{m \cdot \binom{m-1}{|H|}} \varphi_i(Q_H, T_j) \tag{11}$$

i.e. the Owen coalition value is defined on the basis of the Shapley value (extended as in Eq. (9)), similarly to the approach of the so-called "*two-steps value*" formulation of (Owen, 2013, p.300).

*Example* 3. *Consider a coalition structure* $\mathcal{T} = \{\{1,2\}, \{3,4,5\}, \{6\}\}$. *The coalition value* $\Omega_1(\mathcal{T}) = \eta_1(\varnothing, \mathcal{T})$ *is the weighted sums of eight marginals:*

$$\frac{1}{6}\Delta_1(\varnothing) \quad \frac{1}{6}\Delta_1(\{2\}) \quad \frac{1}{6}\Delta_1(\{3,4,5,6\}) \quad \frac{1}{6}\Delta_1(\{3,4,5,6,2\})$$
$$\frac{1}{12}\Delta_1(\{6\}) \quad \frac{1}{12}\Delta_1(\{6,2\}) \quad \frac{1}{12}\Delta_1(\{3,4,5\}) \quad \frac{1}{12}\Delta_1(\{3,4,5,2\})$$

*Since player 1 is in an a-priori coalition with player 2, the other two teams* $\{3,4,5\}$ *and* $\{6\}$ *can only appear as a whole. As a consequence, the Owen approximation of the Shapley coefficients only observes some coalitions, that preserve the integrity of the teams that are in a separate branch of the tree hierarchy.*

Observe that $\Omega_i(\mathcal{T}) \neq \varphi_i(\varnothing, \mathcal{N})$, as only a selected structured subsets of coalitions are formed (see López & Saboya (2009) for an in-depth analysis of this relation).

The two-level formulation is easily extended to an arbitrary hierarchy of coalitions, and this idea has been pioneered for image data by the SHAP Partition Explainer (Lundberg, 2020; Shrikumar et al., 2017; Lundberg & Lee, 2017). Therefore a hierarchical *Owen coalition value* can be obtained rewriting Eq. (11) on top of other Owen coalition values for a coalition $T$, as long as $T$ is not an indivisible coalition. The concept is also briefly sketched in (Owen, 1977, p.87), but we rewrite the equation to have a simple recursive formula that is general for $m$-ary and binary hierarchical coalition structures, as in Eqs. (2) and (3), respectively.

---

[4]In a two-level coalition structure hierarchy $\mathcal{T}$, we have $\mathcal{T} \downarrow = \{T_1 \ldots T_m\}$, and $\forall \, 1 \leq i \leq m$: $T_i \downarrow = \bot$.

binary and multi-way tree hierarchies (i.e. $m > 2$).

Consider Eq. (11) and replace the summation over the subsets of indices $M$ with a uniform *subset $U$ of the sub-coalition structure of $T\downarrow$*, making the marginal contribution of Eq. (1) as the base case of the recursion, and adding a persistent set $Q$ as done for Eq. (9).

$$\Omega_i(Q, T) = \begin{cases} \sum_{U \subseteq T\downarrow \setminus \{T_j\}} \dfrac{1}{m \cdot \binom{m-1}{|U|}} \Omega_i(Q \cup Q_U, T_j\downarrow) & \text{if } T\downarrow = \{T_1 \ldots T_m\} \\ \dfrac{1}{|T|} \Delta_T(Q) & \text{if } T \text{ is indivisible} \end{cases} \tag{12}$$

where $Q_U = \bigcup_{k=1}^{|U|} U_k$, and assuming $T_j$ contains $i$. As before, indivisible coalitions receive uniform attributions among all players. The Owen coalition value for player $i$ using Eq. (12) is obtained from $\Omega_i(\varnothing, \mathcal{T})$, with $\mathcal{T}$ the HCS root. When $\mathcal{T} = \{\mathcal{N}\}, \mathcal{T}\downarrow = \bot$, then Eq. (12) reduces to $\varphi_i(Q, \mathcal{N})$, which is trivially equivalent to Eq. (9). Using a two-level HCS, then Eq. (12) is equivalent to Eq. (10) and Eq. (11). For arbitrary nested hierarchies, the equation expands, generating the coalitions $Q$ that may pair with the set $T$ containing player $i$, following the hierarchy constraints.

*Example 4. Consider a three-level HCS $\mathcal{T} = \Big\{ \{\{1, 2\}, \{3, 4\}\}, \{\{5, 6\}, \{7\}, \{8\}\} \Big\}$. The hierarchical coalition value $\Omega_1(\varnothing, \mathcal{T})$ is the weighted sums of eight marginals:*

$$\begin{array}{llll} \frac{1}{8}\Delta_1(\varnothing) & \frac{1}{8}\Delta_1(\{2\}) & \frac{1}{8}\Delta_1(\{5, 6, 7, 8\}) & \frac{1}{8}\Delta_1(\{5, 6, 7, 8, 2\}) \\ \frac{1}{8}\Delta_1(\{3, 4\}) & \frac{1}{8}\Delta_1(\{3, 4, 2\}) & \frac{1}{8}\Delta_1(\{5, 6, 7, 8, 3, 4\}) & \frac{1}{8}\Delta_1(\{5, 6, 7, 8, 3, 4, 2\}) \end{array}$$

*Coalitions can pair with player $1$ following the hierarchy. Therefore $\{3, 4\}$ and $\{5, 6, 7, 8\}$ can only appear as a whole block from the point-of-view of player $1$, even if the partition $\{5, 6, 7, 8\}$ is not a single coalition.*

Eq. (12) applies to $m$-ary coalition structure, but the case for binary hierarchies is simpler. By assuming $m = 2$, the formula $\Omega_i(Q, T)$ of Eq. (12) can be simplified, obtaining Eq. (4) and completing our derivation.

## A.2 PROOF OF THEOREM 1

Applying Eq. (4) to a partition $T$ that admits a sub-coalition structure $T\downarrow = \{T_1, T_2\}$ creates four branches (two for $i \in T_1$ and two for $i \in T_2$) and necessitates two $\nu$ evaluations. Since we are assuming the BHCS hierarchy to be a balanced tree with depth $d$, we can define the total number $a(d)$ of $\nu$ evaluations for the expansion of all nodes up to depth $d$. Such quantity $a(d)$ follows a linear recurrence sequence represented by Eq. (13):

$$a(d) = \begin{cases} 4 \cdot a(d-1) + 2 & \text{if } d > 0 \\ 0 & \text{if } d = 0 \end{cases} \tag{13}$$

Recursion from Eq. (13) can be eliminated, since the equation is a well-known non-homogeneous linear recurrence with constant coefficients, having solution

$$a(d) = \alpha \cdot a(d-1) + \beta = \frac{\beta(\alpha^{d-1} - 1)}{\alpha - 1}$$

By using $\alpha = 4$ and $\beta = 2$, Eq. (13) simplifies to:

$$a(d) = \frac{2}{3}(4^{d-1} - 1) \tag{14}$$

Thus, the time complexity of Eq. (4) exhibits exponential growth, approximately $O(4^d)$.

## A.3 PSEUDO-CODE OF THE OWEN APPROXIMATION ALGORITHM

A limitation of equation Eq. (4) is that the same coalitions are generated in the recursive expansion of $\Omega_i^{\mathrm{B}}(\varnothing, \mathcal{T})$, for different players $i \in \mathcal{N}$. This issue may severely limit the performance, but it can be easily solved either by memoization, or by generating all the coalitions using a tree visit.

---

**Algorithm 1:** Iterative implementation of Equation Eq. (4).

```
 1  function OwenValues(ν, 𝒯, b)
 2      foreach i ∈ 𝒩  do  Ω^B[i] ← 0
 3      queue.push(⟨1, ∅, 𝒯, ν(∅), ν(𝒩)⟩)
 4      while  queue is not empty do
 5          w, Q, T, v_Q, v_{Q∪T} ← queue.pop()
 6          if T is indivisible or b ≤ 1 then
 7              foreach i ∈ T  do  Ω^B[i] ← Ω^B[i] + w/|T| (v_{Q∪T} − v_Q)
 8          else
 9              T_1, T_2 ← T↓
10              v_{Q∪T_1} ← ν(Q ∪ T_1);  v_{Q∪T_2} ← ν(Q ∪ T_2);   b ← b − 2
11              queue.push(⟨w/2, Q, T_1, v_Q, v_{Q∪T_1}⟩, ⟨w/2, Q ∪ T_2, T_1, v_{Q∪T_2}, v_{Q∪T}⟩
                    ⟨w/2, Q, T_2, v_Q, v_{Q∪T_2}⟩, ⟨w/2, Q ∪ T_1, T_2, v_{Q∪T_1}, v_{Q∪T}⟩)
12      return  Ω^B
```

---

An efficient iterative implementation of the latter is sketched in Algorithm 1, and it is conceptually equivalent to the Partition Explainer of SHAP (Lundberg, 2020). Therefore it does not constitute a novel paper contribution, but we report it for reader's convenience and self-containment.

Algorithm 1 operates at the partition level. It starts from the full coalition at the root $\mathcal{T}$ of the BPT hierarchy (measuring the difference $\nu(\mathcal{N}) - \nu(\varnothing)$). Partitions are inserted into a queue, assumed to be ordered by a priority $w$. It then proceeds by splitting the next partition with the highest $w$, using Eq. (4). Each split requires two model evaluations (line 10), thus reducing the budget $b$ by 2. The splitting continues until the budget $b$ is consumed, or all partitions left are indivisible.

A.4   PSEUDO-CODE OF THE BPT ALGORITHM

Detailed pseudo-code for the BPT algorithm can be found in (Salembier & Garrido, 2000; Randrianasoa et al., 2018; 2021), but a pseudo-code is provided in Algorithm 2. The algorithm is made by three functions:

- **init_bpt**: initializes the unitary partitions $i$ of the BPT hierarchy from the individual pixels $px$ of the input image $x$, and creates the heap of all the pairs of adjacent pixels.
- **get_dist**: computes the distance between two (adjacent) partitions $i$ and $j$ using Eq. (5).
- **build_bpt**: ieratively merges adjacent partitions in *distance*-order, each time creating a new merged partition $k$, and updates the weights in the heap accordingly. The function proceeds as long as there are adjacent partitions, i.e. it stops when all pixels are merged into a single root partition.

Once Algorithm 2 has generated a *merging sequence*, it can be efficiently stored into 6 arrays:

- $leaf\_idx[i]$: the image pixel of unitary coalition $i$, with $i \in [1, n]$;
- $left\_branch[k]$ and $right\_branch[k]$: the two partition indexes resulting from the split $T_k\downarrow$ of each non-unitary coalition $k$, with $k \in [n+1, 2n-1]$;
- $start[k]$ and $end[k]$: the index interval of pixels for the non-unitary partition $k$;
- $pixels$: the sorted array of pixel indexes, indexed by $start$ and $end$.

Therefore, the space needed to store the BPT hierarchy in memory is $\Theta(6n)$ integers.

The core data structure is a graph of the partitions (nodes), paired with the list of adjacencies (edges). The adjacency list needs to be sorted efficiently in order to extract the edge $adj = (i, j)$ having the smallest $dist(i, j)$, as defined by Eq. (5) and computed by function **get_dist**. To do so, a heap data structure is a reasonable choice. Merging coalitions therefore requires to both modify the nodes and update the edges. This process, described at line 11 of **build_bpt** and depicted in Figure 2/B, shows that each merge operation requires to traverse the adjacency list of the merged partitions. Further details are provided in the paper of Randrianasoa et al. (2018).

**Algorithm 2:** Pseudo-code of the BPT algorithm.

```
1  function init_bpt(𝒳:image)
2      foreach pixel px of image x do
3          i ← make_partition()
4          minR[i] ← maxR[i] ← R[px]
5          minG[i] ← maxG[i] ← G[px]
6          minB[i] ← maxB[i] ← B[px]
7          area[i] ← 1;  perimeter[i] ← 4;  root[i] ← i
8      foreach pair of partitions i, j that are adjacent pixels in x do
9          heap_push(heap, make_adjacency(i, j, weight=get_dist(i, j)) )
```

```
1  function get_dist(i, j)
2      rangeR ← max(maxR[i] − maxR[j]) − min(minR[i] − minR[j])
3      rangeG ← max(maxG[i] − maxG[j]) − min(minG[i] − minG[j])
4      rangeB ← max(maxB[i] − maxB[j]) − min(minB[i] − minB[j])
5      area ← area[i] + area[j]
6      perimeter ← perimeter[i] + perimeter[j] − 2 * adjacent_perimeter[i, j]
7      return  (rangeR² + rangeG² + rangeB²) * area * √perimeter
```

$$\text{return } (rangeR^2 + rangeG^2 + rangeB^2) * area * \sqrt{perimeter}$$

```
1  function build_bpt()
2      while heap is not empty do
3          adj ← heap_pop(heap)
4          i, j ← partitions in adj;  k ← make_partition()
5          minR[k] ← min(minR[i], minR[j]);  maxR[k] ← max(maxR[i], maxR[j])
6          minG[k] ← min(minG[i], minG[j]);  maxG[k] ← max(maxG[i], maxG[j])
7          minB[k] ← min(minB[i], minB[j]);  maxB[k] ← max(maxB[i], maxB[j])
8          area[k] ← area[i] + area[j];  perimeter[k] ← perimeter[i] + perimeter[j]
9          root[k] ← k;  root[i] ← root[j] ← k
10         left_branch[k] ← i;  right_branch[k] ← i
11         merge linked lists of adjacencies of i and j into a single linked list for partition k, updating the heap
              weights using get_dist since partitions i and j are now merged together.
```

## A.5  PYTHON IMPLEMENTATION

A Python implementation, named *ShapBPT*, is provided. A snippet of the python code using the *ShapBPT* module to obtain a Shapley explanation for a given image using the masking function $\nu$ is provided in Algorithm 3. While not detailed in the paper, the implementation supports multi-class explanations, similarly to (Lundberg, 2020).

**Algorithm 3:** Example Python code.

```python
1  from shap_bpt import Explainer
2  explainer = Explainer(ν, image_to_explain, num_explained_classes)
3  shap_values = explainer.explain_instance(max_evals=b)
```

## A.6 ADDITIONAL SALIENCY MAPS FOR EXPERIMENT E1

Figure 6 shows additional saliency maps for the **E1** experiment, generated by explaining the classification of the ResNet50 model on the samples from the ImageNet-S$_{50}$ dataset.

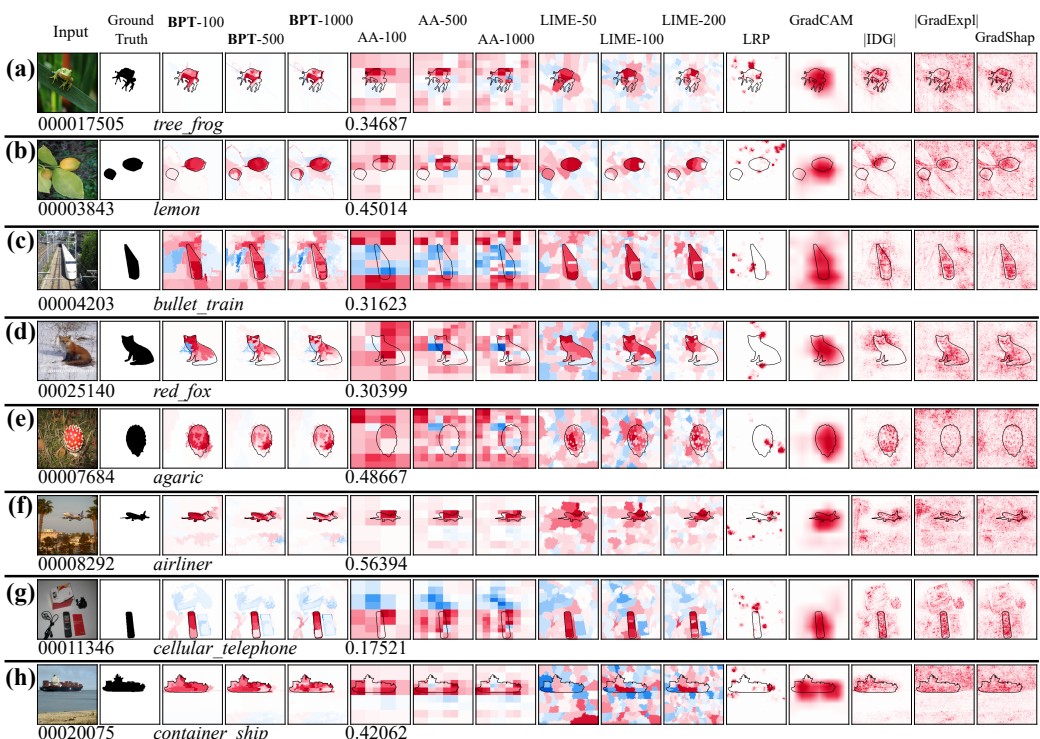

Figure 6: Additional saliency maps generated for the **E1** experiment.

| E1: 574 images, 14 methods. | | |
|---|---|---|
| **Metric** | **$p$-value** | **Pairs not rejecting $H_0$** |
| $AUC^+$ | 0.0 | - |
| $AUC^-$ | $9.22 \times 10^{-186}$ | - |
| $MSE^+$ | $87559 \times 10^{-3}$ | - |
| $MSE^-$ | $7.55 \times 10^{-3}$ | - |
| $max\text{-}IoU$ | $1.20 \times 10^{-197}$ | - |
| $AU\text{-}IoU$ | 0.0 | - |

Table 2: Summary of the one-way ANOVA test for the **E1** experiment.

## A.7 EXPERIMENT E2

One important limitation of experiment **E1** is that the ground truth may not be faithful, as the black-box model may classify an object based on partial details or using weak correlations. To overcome this limitation, we repeat the experiment adopting an ideal model which perfectly follows the ground truth. Let

$$\nu_{\text{lin}}(S) = \frac{|S \cap G|}{|G|} \tag{15}$$

be an ideal linear model that outputs the proportion of pixels of $S$ that belong to the ground truth $G$. Since $\nu_{\text{lin}}$ is not a neural network, CAM methods cannot be used and are excluded. To better compare BPT and AA, we also add two other AA variations, with a budget of 5000 and 10000 samples. By using a linear model, the experimental environment has minimal noise, is therefore simpler to interpret, and provides a better baseline for assessment, even if it is less realistic than a deep learning model.

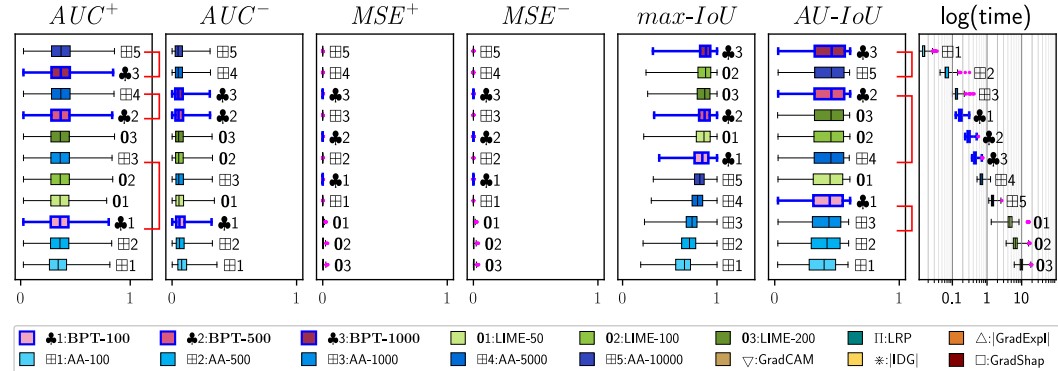

Figure 7: Results for the six metrics across 574 images from the ImageNet-S$_{50}$ dataset, with methods sorted to display the highest-performing one atop each column, for the experiments **E2**. Higher is Better for $AUC^+$, and $max - IoU$ and Lower is Better for the remaining ones.

Figure 7 shows the results of experiment **E2**, while a subset of the generated saliency maps are depicted in Figure 8. The results shows the effectiveness of the BPT explanation strategy: all BPT-$b$ achieve better scores that their AA-$b$ counterpart, for the same budget $b$. Interestingly we observe that, in terms of both $AUC^+$ and $AU$-$IoU$, the BPT strategy achieves comparable scores to the AA strategy while employing only a tenth of the evaluation samples (relations highlighted by red brackets in Figure 7).

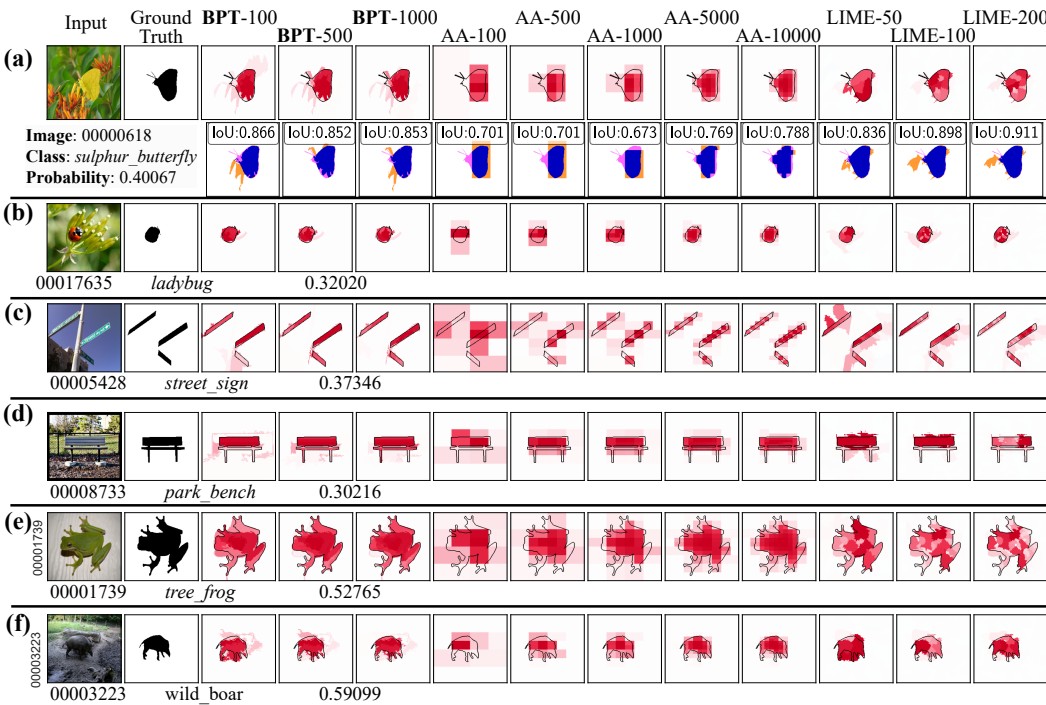

Figure 8: Saliency maps obtained from the ideal linear model $\nu_{\text{lin}}$.

## A.8   EXPERIMENT E3

We also consider a third experimental setup **E3**, whose results are depicted in Figure 9. In this setup, the masking function $\nu(S)$ is defined as the average of the model evaluation when using multiple replacement values instead of a single one. The considered replacement values are: I. gray value (0.5); II. black value (0.0); III. white value (1.0); IV. Gaussian noise, with average 0.5; V. input image passed through a Gaussian blur filter with kernel size of 8. The limit of using a single replacement strategy/value is that an image region may be replaced with a value that is close to the original one. By using multiple different replacement values, such risk is reduced, and the obtained values can be expected to be more robust. The limit is that an evaluation of $\nu(S)$ for a set $S$ now requires multiple evaluations of the explained model $f(x)$.

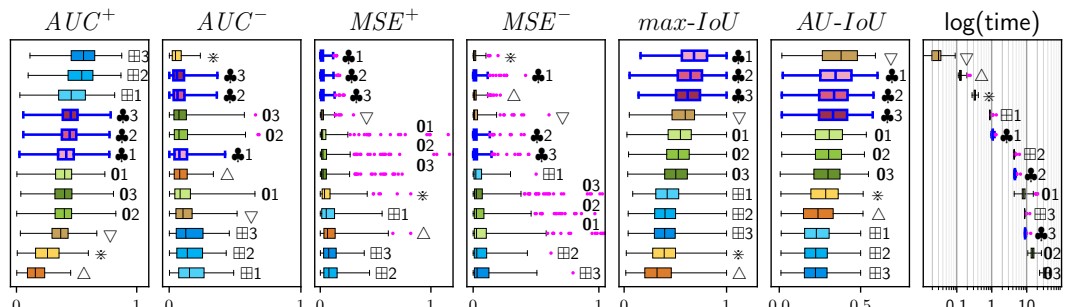

Figure 9: Results of the experimental setup **E3** using five replacement values. Higher is Better for $AUC^+$, and $max - IoU$ and Lower is Better for the remaining ones.

Even if we consider five replacement values instead of just one, the results in **E3** for the BPT, AA and LIME methods remains similar to the ones of **E1**. Again, BPT stands close to the top of most scores, and it always surpasses AA except for $AUC^+$.

The case of $AUC^+$ is very interesting and revealing. We empirically observed that, in several instances, the behavior of AA resembles that depicted in Figure 3(B). In this case, the model classifies the input $x$ as the class *indigo_bunting* with a probability of 0.444. As pixels are added in decreasing Shapley order, the BPT explanation reaches approximately 0.444 and remains stable as background pixels are included (red curve). Conversely, the AA explanation exhibits a significant overshoot: the probability increases above 0.444 and then gradually decreases (blue curve). We observed this behaviour also in **E1** and **E3** experiments. Although this behavior yields higher *area-under-curve* scores, we suspect that the expected behavior should align with the former, not the latter. Further investigation is required in this area.

## A.9   EXPERIMENT E4

All evaluations in experiments **E1** and **E3** were conducted using the ResNet50 model. While the proposed strategy is model-agnostic, it is nonetheless interesting to observe its behaviour with different deep learning model architectures. In experiment **E4** we replicate the same setup of **E1** but using the VGG-16 model of Simonyan (2015), using the pretrained *1K_V1* weights found in the pytorch library that have 90.382% Top-5 accuracy. Numerical results are reported in Figure 11, and a subset of the generated saliency maps is depicted in Figure 10.

As for the previous experiments, the BPT strategy shows top scores in almost all the tested scores except one ($AUC^-$)

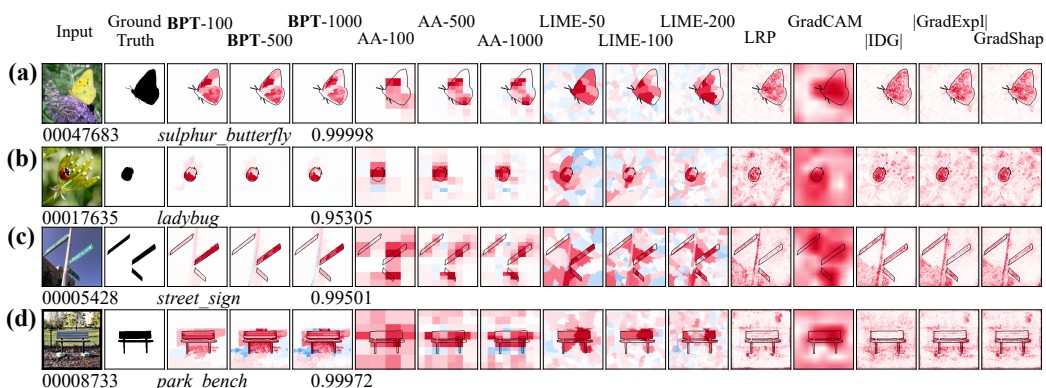

Figure 10: Saliency maps from selected instances in the **E4** experiment (using VGG-16 model).

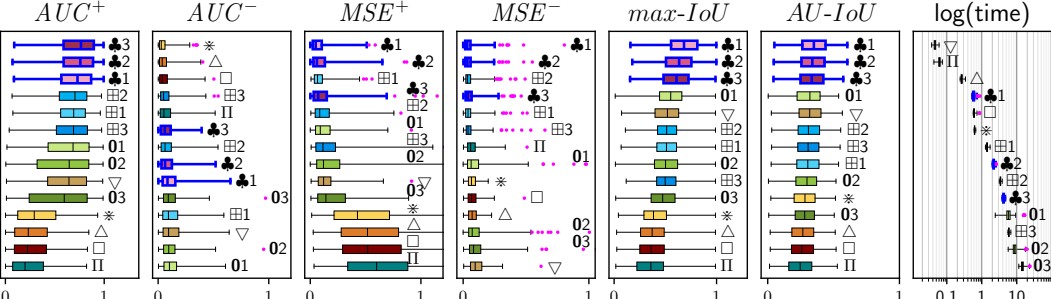

Figure 11: Results for the six metrics across 525 images from the ImageNet-S50 dataset, with methods sorted to display the highest-performing one atop each column, for the experiments **E4**. Higher is Better for $AUC^+$, and $max - IoU$ and Lower is Better for the remaining ones.

## A.10 EXPERIMENT E5

Similarly to experiment **E4**, we also tested the proposed method on Vision Transformer models. We selected the Swin-ViT model of Liu et al. (2021), and the summary of the results is shown in Figure 13. Again, a few saliency maps from the same set of selected examples is also shown in Figure 12.

The LRP method implementation we used does not support this transformer model architecture, therefore we excluded it from the results. As a first observation, it is interesting to see that all methods except BPT produce significantly more confused explanation, attributing a lot of importance to background features and with little focus to the actual classified objects. On the contrary, saliency maps obtained by the BPT method are more clear and focused. Again, BPT seems to excel in all scores, being surpassed on *MSE* scores by a small margin only by AA.

This experiment is particularly revealing, as ViT models appears to be more robust at input masking, and are therefore more difficult to explain using model-agnostic methods (w.r.t. convolutional models) that require feature replacement to probe the model behaviour.

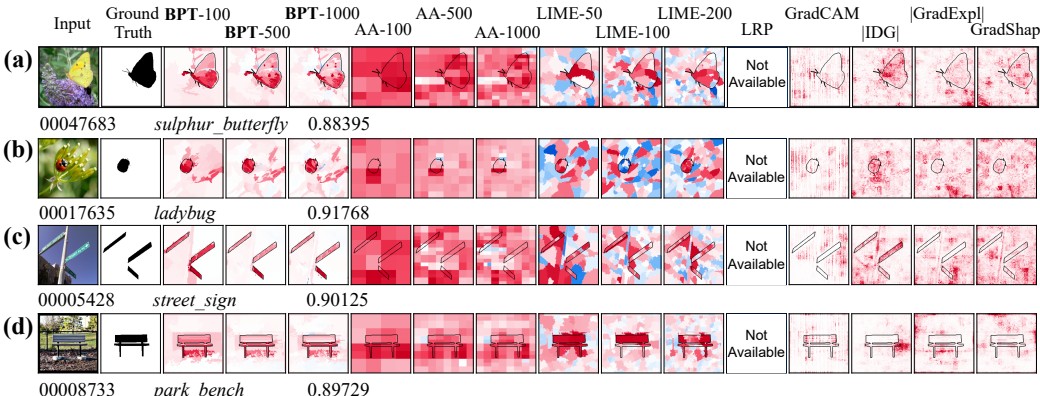

Figure 12: Saliency maps from selected instances in the **E5** experiment (using Swin-ViT model)

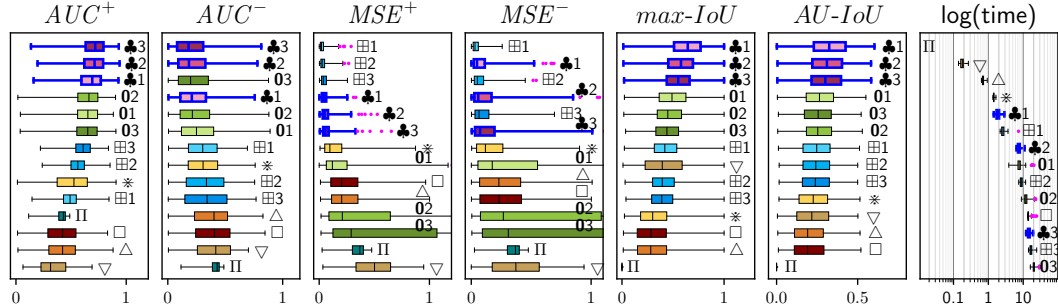

Figure 13: Results for the six metrics across 621 images from the ImageNet-S50 dataset, with methods sorted to display the highest-performing one atop each column, for the experiments **E5**. The explained model is the Swin-ViT vision transformer model. Higher is Better for $AUC^+$, and $max - IoU$ and Lower is Better for the remaining ones.

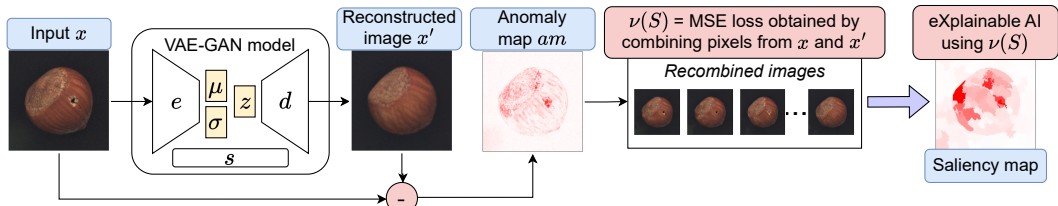

Figure 14: Workflow of the explainable AI applied to the Anomaly Detection system of **E6**.

## A.11    EXPERIMENTS E6

All the results presented so far are variations of the ImageNet classification challenge. However, given the broad applicability of explainable AI to different practical problems, it is also interesting to see how it behaves in other settings. For experiment **E6** we consider the problem of explaining anomalies detected by an Anomaly Detection (AD) system on image data. This experiment is based on the work of Ravi et al. (2021) where anomalies in images are detected using a Variational AutoEncoder-Generative Adversarial Network (VAE-GAN) model by means of anomaly localization. We use the MVTec benchmark dataset (Bergmann et al., 2019) which has 5000 high quality images with defective and non-defective samples from 15 different categories of objects. We selected the *hazelnut* object category from the dataset.

The pipeline of this system is depicted in Figure 14. An input image $x$ is reconstructed into $x'$ using a one-class VAE-GAN classifier. The anomaly map $am$ captures the reconstruction error, which sums up both the potential anomalies of $x$ as well as the noise. An XAI method can be employed to separate the noise from the detected anomalies, thus localizing if and where the anomalies are present. In this contest, the function $\nu(S)$ is a MSE loss on the anomaly map $am$ itself. Since $\nu(S)$ is not a neural network, we cannot use CAM methods. Therefore, we generate saliency maps using BPT, AA and LIME. We use values 100, 500, and 1000 for the budget value $b$. For LIME, we use 50,100 and 200 a-priori segments, respectively.

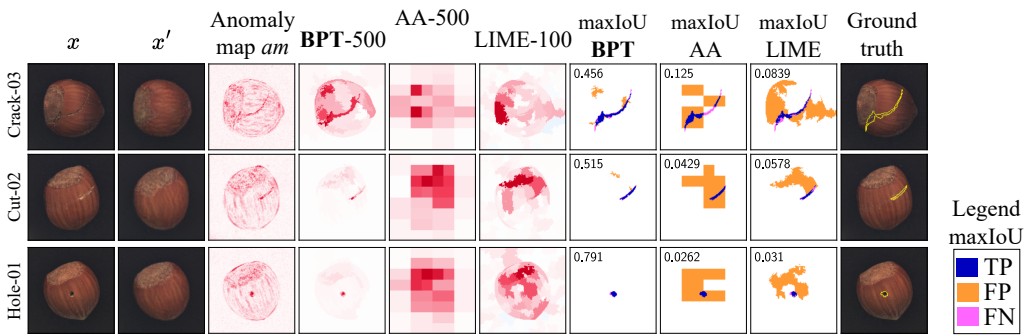

Figure 15: Selected examples in the Anomaly Detection system for experiment **E6**.

As the MVTech dataset has proper ground truth masks for the expected anomalous regions, we can compute all the six scores defined in Section 4. Figure 15 shows the AD problem on three input images. For each input, a row shows: the input $x$, its reconstruction $x'$ through the VAE-GAN model, the anomaly map $am$, the explanation generated by BPT with $b$=500, by AA with $b$=500 and by LIME with $b$=500 and 100 segments. The best intersection-over-union is also shown, highlighting the True Positives (TP), the False Positives (FP) and the False Negatives (FN). The ground truth $g$ is also shown, for reference.

Results are reported in Figure 16. Again, all three XAI method are capable of identifying the real anomalous regions on the various samples, but BPT significantly outperforms the others. This is particularly true for the task of identifying the exact region, which is highlighted by the very high *max-IoU* scores.

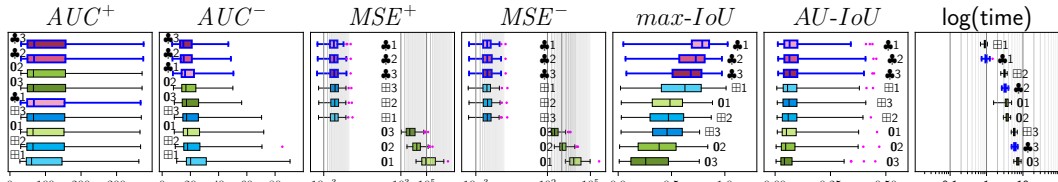

Figure 16: Results for the six metrics with methods sorted to display the highest-performing one atop each column, for the experiments **E6**. Higher is Better for $AUC^+$, and $max-IoU$ and Lower is Better for the remaining ones.

## A.12   EXPERIMENTS E7

As a last experiment, we consider a third setting that adopts a multiclass regression model instead of a classification model. The goal is to determine the presence (positive prediction) or absence (negative prediction) of a given facial feature, like *brown-hair* of *eye-glasses*, while the XAI task consists in localizing the regions that drive such prediction. The dataset is CelebA-HQ (Karras et al., 2018). Among the 40 attributes, we tested two attributes *brownhairs*, and *eyeglasses* whose ground-truth could also be established from a segmentation mask. This results in 106 images tested. We use a pre-trained sequential CNN model, provided by (Batra, 2020). An example of the XAI task is shown in Figure 17. Three instances are shown: (a) a subject with brown hair, who is recognized having *brown-hair* (score is positive); (b) a subject with black hair, who is recognized not having *brown-hair* (score is negative);; (c) a subject wearing eyeglasses who is recognized having them. For case (a) and (c), Shapley values are positive in the areas that drive the positive score. Conversely, for case (b), Shapley values are negative in the areas that drive the negative score. CAM methods do not have this property (as they are not Shapley values and do not obey the efficiency axiom), so we take them in absolute value.

Results of the evaluation are reported in the tables in Figure 18. This experiment shows again the capacity of BPT-based methods to adaptively follow the borders of the activating regions, achieving high performances particularly on IoU scores. Note that also in this case, as previously discussed for **E1**, the ground truth can only be considered as a weak approximation of the model's learnt representation, as the model is likely to use multiple features of the subject face to determine the presence or the absence of a specific attribute, not just the shape of the hair or the eyeglasses. Nonetheless, the localization of that area remains more precise when data-awareness is used.

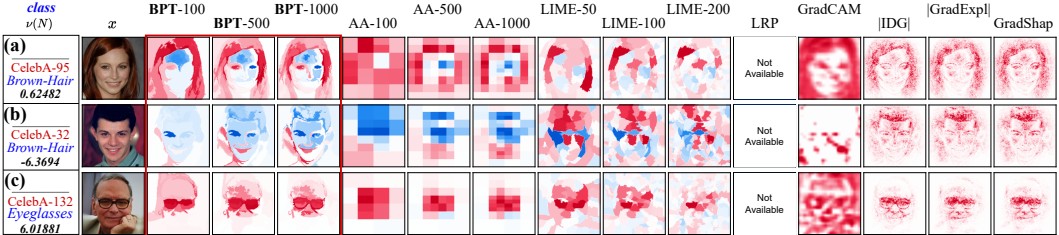

Figure 17: Examples of the **E7** experiment, explaining facial attributes using the CelebA dataset.

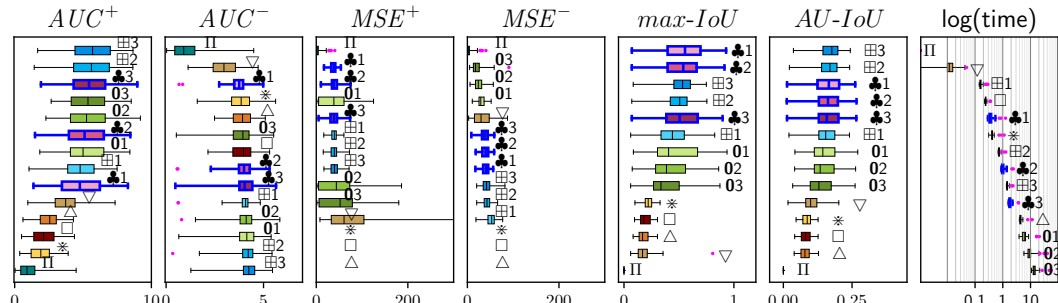

Figure 18: Results for the 6 metrics across selected images from the CelebA dataset, with methods sorted to display the highest-performing one atop each column, for the experiments **E7**. Higher is Better for $AUC^+$, and $max - IoU$ and Lower is Better for the remaining ones.

## A.13 INSTANCES WITH MULTIPLE OBJECTS

The proposed BPT algorithm is not limited to identifying single objects or contiguous regions. The recursive expansion in Eq. (4) does not constrain the regions with the highest Shapley value distribution mass to be contiguous. To illustrate this, we provide additional examples of such cases within the context of the ImageNet-$S_{50}$ dataset, experiment **E1**, reported in Figure 19. Of course, the actual capacity of finding multiple separate objects depends also on the black box model $f$ being able to detect them separately.

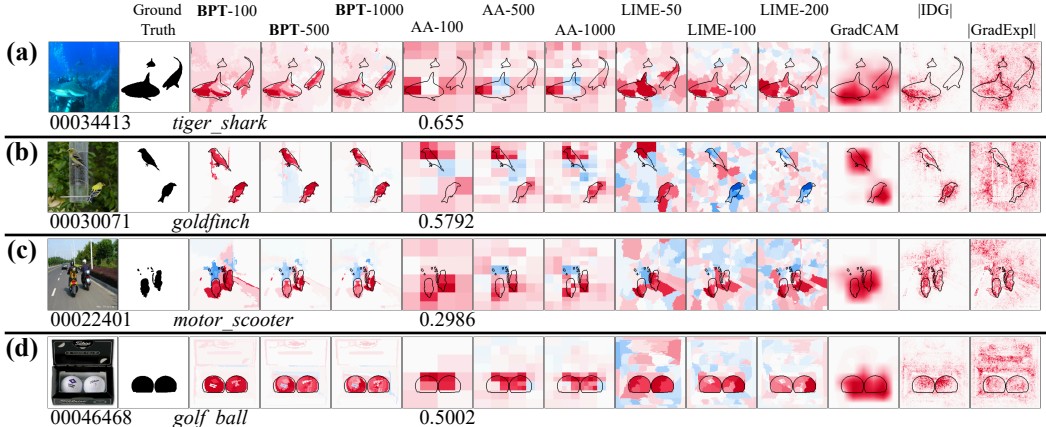

Figure 19: Selected examples in experiment **E1** with multiple objects.

## A.14 RANDOMIZATION TEST

In all previous experiments, we assume that the BPT hierarchy is constructed using the same input image, making it inherently data-aware. However, a question not addressed in these tests is: *What happens if the target object(s) is not well-aligned with such a hierarchy?* To evaluate performance in this scenario, we designed a *randomization test*. In this test, the BPT hierarchy is constructed using a different image than the one being explained, resulting in a hierarchy that is not aligned with the actual data being explained. We refer to this "misaligned" BPT hierarchy as **BPT**R (Random). The goal is to compare the performance of BPTR against a baseline (AA), assessing whether the results remain at least comparable despite the misalignment.

Figure 20 shows on the first line the image used for constructing the BPT hierarchy, used then to generate the explanations (in the **E2** context) for the other three sample images (a), (b) and (c) in the BPTR columns. Results are compared against the standard data-aware BPT, and the AxisAligned strategy of SHAP.

**(mask)**

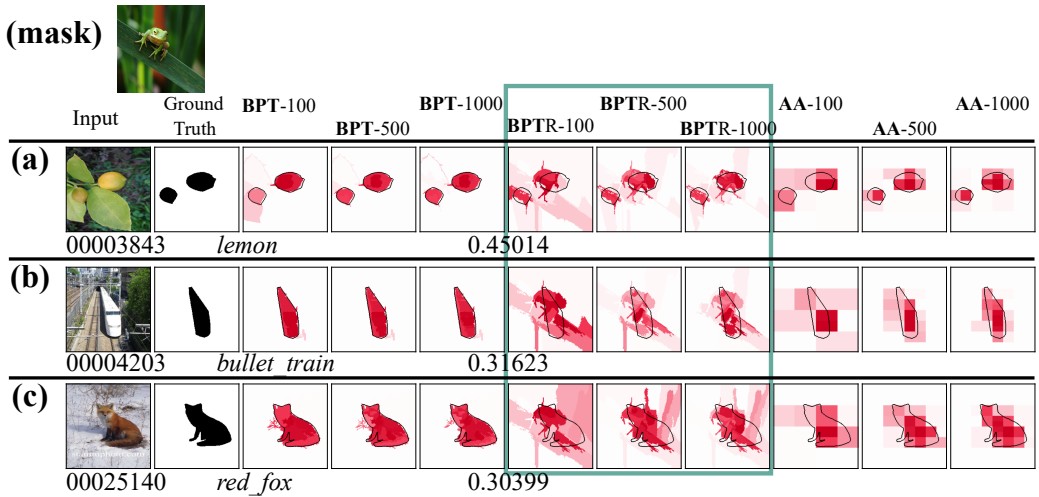

Figure 20: Randomization test for experiment **E2** where BPTR hierarchies are computed from an unrelated image **(mask)**.

Results are TBD.

