# OpenReview forum: "Shapley Image Explanations With Data-Aware Binary Partition Trees"
_ICLR.cc/2025/Conference — Submitted to ICLR 2025_

### Official Review · Reviewer_EuK2 · 2024-10-28

**Soundness:** 2
**Presentation:** 3
**Contribution:** 2
**Rating:** 3
**Confidence:** 4

**Summary:**

This paper proposes an XAI method which combines the SHAP method with a Binary Partition Tree (BPT) representation of an image. The BPT is used to segment the image into regions which are masked to determine their contribution to a model’s classification decision. The proposed method is evaluated by comparing the selected “important” regions to the ground-truth object segment in an object classification task. The proposed method outperforms methods which segment the image into superpixels or grids.

**Strengths:**

- The paper is clearly written and the results are well-presented. In particular, the figures are well designed to illustrate how the method works.
- The paper does include an experiment where the model's ground truth is known (E2), which is something that's lacking in a lot of XAI research. I have some concerns with this experiment (the chosen ground truth seems to strongly favor this model over the comparison models), but it's still nice to see this kind of evaluation in an XAI paper.

**Weaknesses:**

- The main novelty seems to be using BPT instead of a simpler image segmentation like a grid or superpixels.
- The method is only compared to methods that use these simpler segmentation approaches. It's not surprising that this approach performs better.
- The “ground truth” explanation region used in the evaluations is a human-annotated semantic segmentation of the object (Gao, et al., 2022). This probably isn’t the actual ground truth region for a classification model -- it's likely a model focuses on just a few parts of the segment that are most critical for recognition -- but this choice of ground truth seems to clearly favor a model that can produce object-like segments over models that only use very simple segments like grids or superpixels. E2 seems to be intended to address this issue, but E2 also uses semantic segments as the ground truth "important" region for the model. It would be more convincing to show that this model still work better than other approaches when the ground truth is not a semantic segment.

**Questions:**

- How well does this method work when the "ground truth" important region is not a whole object or whole semantic segment (what if the critical feature is an edge or corner or just part of an object)? This approach seems to assume that connected same-color regions are the same thing and contribute equally to importance, and it's only tested with data that matches this assumption.
- Why is it not possible to swap out BPT with other segmentation methods? The paper says that "fixed" segmentations cannot be used but most segmentation methods allow hierarchical merging; it wasn't particularly clear to me why none of them are applicable in approach. It does seem like a potential limitation of this approach that the user cannot use alternate segmentation algorithms.

---

> ### Author Response · Authors · 2024-11-19
>
> ### Weakness 1
> A data-aware hierarchy compatible with the Owen formula has not been proposed before, and we believe it represents a significant and novel contribution. Its impact is demonstrated both qualitatively and quantitatively through our extensive experimental evaluation.
>
> ### Weakness 2
> We compared ShapBPT against both trivial (AA) and advanced segmentation algorithms, such as LIME, which utilizes *quickshift*. The superior performance of BPT over LIME highlights the advantage of integrating hierarchy exploration within the recursive Owen formula. This integration is a non-trivial enhancement compared to existing state-of-the-art XAI methods.
>
> ### Weakness 3
> We addressed the limitations of the ground truth in lines 418–424. Additionally, we included response curve-based metrics (AUC and MSE) in our evaluation, which are independent of any ground truth.
>
> Regarding the last point, we can design an additional **randomization experiment** where the BPT hierarchy is generated using a different image (instead of the one being explained). The goal of this experiment is to test that, even in a scenario where the hierarchy is poorly aligned (or not aligned at all) with the data, the results remain comparable to those obtained using Axis-Aligned segmentation in SHAP.  A sketch of this test is added in the PDF, in Appendix A.14, pages 23-24.
>
> ### Question 1
> The use of ground truth in our study is primarily within the context of the *Visual Recognition Challenge (VRC)* as defined by [Russakovsky2015]. This involves object-level annotations in the form of binary masks (like the one provided by the ImageNet-S50 dataset [Gao2022]). Our ground-truth-based evaluation is limited to the VRC case, as it is both well-studied and widely recognized.
>
> To complement this, we also provided response curve metrics (AUC and MSE), which do not rely on ground truth. These latter metrics align with the context you are highlighting, offering a broader evaluation framework.
>
> ### Question 2
> The highly efficient Owen approximation formula requires a BHCS with the specific characteristics outlined in lines 188–203. Most segmentation algorithms do not fully meet these requirements. While hierarchical merging techniques could potentially be used, we opted to focus on a clear and direct comparison between BPT and well-established state-of-the-art methods, such as Axis-Aligned SHAP and LIME with quickshift.
>
> Future work could explore hybrid approaches or alternative hierarchical methods, but this study prioritizes presenting a coherent proposal based on BPT integrated with the Owen formula.
>
> *[Russakovsky2015] Olga Russakovsky, Jia Deng, Hao Su, Jonathan Krause, Sanjeev Satheesh, Sean Ma, Zhiheng Huang, Andrej Karpathy, Aditya Khosla, Michael Bernstein, Alexander C. Berg, and Li Fei-Fei. ImageNet Large Scale Visual Recognition Challenge. International Journal of Computer Vision (IJCV), 115(3):211–252, 2015.*
>
> *[Gao2022] Shanghua Gao, Zhong-Yu Li, Ming-Hsuan Yang, Ming-Ming Cheng, Junwei Han, and Philip Torr. Large-scale unsupervised semantic segmentation. TPAMI, 2022.*

---

### Official Review · Reviewer_ZEi3 · 2024-11-03

**Soundness:** 3
**Presentation:** 4
**Contribution:** 3
**Rating:** 6
**Confidence:** 4

**Summary:**

This paper presents an explainable AI (XAI) method that combines the SHAP partition explainer with the partition hierarchy of the BPT algorithm. While the idea of integrating these two algorithms is straightforward, it results in greater focus on the object region compared to previous methods, as evidenced by numerical experimental results.

**Strengths:**

+ The technical details are carefully and thoroughly described.
+ Although the proposed method simply combines SHAP and BPT algorithms, it shows great promise based on the qualitative comparison in Figure 4 and the quantitative comparisons in Figures 4 and 5. This combination is novel in the context of XAI and demonstrates effective results.
+ All code and notebooks required to reproduce the paper's results are provided and will be made publicly available upon acceptance, facilitating readers’ understanding of the method. Additionally, the proposed XAI method can be immediately implemented by users.
+ The discussion of alternative candidate modules for each stage in Section 5 further enhances understanding of the approach presented in this paper.

**Weaknesses:**

- I have a concern about the ground truth in Figure 4. It seems that the authors are claiming that high scores should be assigned to the target object only and no score should be assigned to any other region, with which I do not fully agree. In some cases, the background information or high-order interactions between the object and the background should also be important and considered. However, in the definition in this paper, such cases are eliminated. Here are my questions or requests:
  - The authors might want to justify their “Truth.”
  - The authors might want to discuss the limitations of their ground truth definition and consider how their method might be evaluated in cases where background or contextual information is important for classification.
  - The authors might want to discuss using alternative ground truth definitions that account for these factors.

- It seems there is a strong assumption: there is only one object in the target image. Here, I have some questions to the authors
  - What kind of responses does the proposed model would yield if there are multiple objects such as two or three butterflies in the image in the case of Figure 4(a)?
  - What would happen if this model is applied to subjective classification tasks such as good/bad quality photos or to regression tasks such as image aesthetics score predictions?

- The choice of the BPT algorithm should be justified. For instance, applying the segment anything model (SAM) or bilateral filter-based region segmentation in the pre-processing stage would also be possible rather than using the BPT algorithm. It would be helpful if the authors could add a more detailed explanation why the BPT algorithm was employed or additional experiments.

Here are some comments to improve the paper (not considered for the paper score):
- There are many mistakes such as undefined abbreviations, grammatical errors, inconsistent reference styles, etc.
- x in Section 2 is preferred to be a bold face because it is a vector.
- Figure 2 is referred to only from the appendix. Therefore, the authors might want to reconsider if this figure is really needed here. Or, if this figure is important to convey the concept of BPT, the authors might want to add more detailed description with proper reference in the main body.
- The authors might want to better illustrate Figure 5 because the current figure is too busy and the values are not readable.

**Questions:**

Please answer my comments and questions in the Weakness part.

---

> ### Author Response · Authors · 2024-11-18
>
> ### Weakness 1
> We define the ground truth following the *Visual Recognition Challenge (VRC)* framework outlined in [Russakovsky2015], utilizing object-level annotations in the form of binary masks provided by ImageNet-S50 [Gao2022]. While we recognize the limitations of this binary ground truth, as discussed in lines 418–424, our evaluation is restricted to the VRC case, which is both well-studied and widely acknowledged.
>
> For non-VRC problems, where defining ground truth is inherently more challenging, we would rely solely on response curve-based metrics (e.g., AUC or MSE). To address this, we can add a discussion in Section 5 about the applicability of ShapBPT to non-VRC problems, noting that this challenge is common to all saliency-based XAI methods.
>
>
> ### Weakness 2
> No, our approach does not rely on any such strong assumptions. This is demonstrated by examples such as Fig. 4(c) (three street signs) and Fig. 6(b) (two lemons). In these cases, the saliency maps effectively represent Shapley values distributed across multiple distinct regions, which aligns with the expected behavior.
>
> That said, subjective classification tasks are inherently more difficult to interpret using saliency maps. As such, our work focuses on problem classes (e.g., VRC) where ShapBPT is most applicable and effective.
>
>
> ### Weakness 3
> We considered SAM in our initial evaluations, but it was ultimately excluded from the experimental setup because it may generate overlapping regions, making it incompatible with the Owen formula (Eq. 12). Instead, we included comparisons with *quickshift* segmentation and incorporated LIME among the benchmarked methods.
>
> The primary reason for adopting BPT with the Owen formula is that it is the only algorithm we are aware of that naturally satisfies both requirements R1 and R2, as detailed in lines 188–203. Furthermore, we believe the positive properties of BPT—such as multi-scale hierarchy and quasi-balanced binary HCS—are not only advantageous for ShapBPT but also beneficial for other machine learning tasks.
>
>
> *[Russakovsky2015] Olga Russakovsky, Jia Deng, Hao Su, Jonathan Krause, Sanjeev Satheesh, Sean Ma, Zhiheng Huang, Andrej Karpathy, Aditya Khosla, Michael Bernstein, Alexander C. Berg, and Li Fei-Fei. ImageNet Large Scale Visual Recognition Challenge. International Journal of Computer Vision (IJCV), 115(3):211–252, 2015.*
>
> *[Gao2022] Shanghua Gao, Zhong-Yu Li, Ming-Hsuan Yang, Ming-Ming Cheng, Junwei Han, and Philip Torr. Large-scale unsupervised semantic segmentation. TPAMI, 2022.*

---

> > ### Comment · Reviewer_ZEi3 · 2024-11-19
> >
> > I appreciate the authors’ detailed responses.
> > Let me clarify that I am still positive about this paper, but I would like to ask questions to make this paper more attractive.
> >
> > - Weakness 1
> >   - I understand that the authors are based on a well-established dataset. However, my concern was not addressed well because the Visual Recognition Challenge (VRC) in [Russakovsky2015] is rather simple: no consideration on the background information or high-order interactions is needed. The authors might want to apply their method to more complicated image datasets such as (just an example) MS-COCO.
> >
> > - Weakness 2
> >   - I admit that I missed the results such as Fig. 4(c) (three street signs) and Fig. 6(b) (two lemons), but I see that little attention is paid to the smaller lemon in Fig. 6(b). It would be helpful if the authors could provide more case studies.

---

> > > ### Author Response · Authors · 2024-11-19
> > >
> > > ### Weakness 1
> > > Apologies for misunderstanding your initial point. We can prepare an additional experiment targeting the MS-COCO dataset, which features larger images with more complex target objects, and report the results. However, we are concerned that there may not be enough time to prepare these results before the end of this Rebuttal Period. Nonetheless, we will ensure they are included in the final version of the paper.
> > >
> > > ### Weakness 2
> > > We have added a new set of multi-object images in Figure 19 (page 23) to illustrate additional cases in the E1 setting where multiple objects are present. Of course the detection of these objects depends on two factors: the ability of the black-box model to identify separate objects and the capability of the XAI method to localize them within distinct, non-overlapping regions.  If the model does not "see" the objects (as in the case of the two little lemons), the XAI method will not identify their regions.

---

### Official Review · Reviewer_iXdf · 2024-11-03

**Soundness:** 2
**Presentation:** 1
**Contribution:** 2
**Rating:** 6
**Confidence:** 3

**Summary:**

The paper discusses about aligning morphological features of an image to enhance the identification of relevant regions for extracting a visual interpretation of a machine learning model for XAI. The authors try to mitigate the poor alignment of pixel level attribution and the morphological features of an image and propose a method to compute the Owen's approximation of Shapley coefficients following a hierarchical coalation structure from BPT.

**Strengths:**

1) Discusses an important aspect of XAI i.e. poor alignment of pixel level attribution and morphological features.
2) Shows experiments with several settings.
3) Considers a wide range of metrics.
4) Discusses about 5 different replacement values (although in Supplementary) and evaluates the same.
5) Provided code in the form of library for reproducibility.

**Weaknesses:**

1) The paper's organization could be clearer, and it isn't easy to understand, especially the section on experimental assessment.

2) For the reader's benefit, it would be nice to put some markers expressing whether the metrics' higher or lower value is better or worse. For example, in the captions of Figures 5, 7, 9, etc.

3) The authors talk about the ANOVA test (in lines 403-409) but don't refer to any table (in the paper or supplementary) related to the test. Please include a table in supplementary or write about the range of p-values so that the readers can get some idea.

4) The evaluation was done on 574 images, which may have needed more. It would be great if the authors could cite previous work using similar dataset sizes for similar evaluations.

5) In Figure 5, specific methods (e.g., IDG in AUC-, GradCAM in AU-IoU) appear to outperform the proposed approach, but the distributions seem to have high overlap. For the benefit of readers, it would be helpful to include statistical test (like Wilcoxon signed rank test or paired t-test) to provide a clearer understanding of whether the observed differences are statistically significant or not.

**Questions:**

The paper discusses an important aspect, namely, aligning pixel-level attribution with morphological features of an image, and that's why I am leaning towards mild acceptance. However, I would like to know the authors' comments regarding the points I mentioned in the weakness section.

---

> ### Author Response · Authors · 2024-11-18
>
> ### Weakness 1
> Could you specify which aspects you found difficult to follow or understand? If you provide more details, we can reorganize the experimental assessment to address your concerns effectively.
>
> ### Weakness 2
> It’s true that different scores have different interpretations, but all plots are sorted to display the highest-performing method at the top (as noted in lines 391–392). To enhance clarity, we can add a visual marker to reinforce this sorting.
>
> ### Weakness 3
> We will include tables containing all the one-way ANOVA p-values in the supplementary material. For reference, the one-way ANOVA p-values for E1 have already been added to the revised PDF (see Table 2, page 16).
>
> ### Weakness 4
> We used all the images available in the ImageNet-S50 dataset (lines 366–370), which is a high-quality, manually annotated subset derived from ImageNet. A similar experimental setup can be found in [Teneggi2022], which used 120 and 300 large images in its experiments. These images came from malaria and traffic light datasets and featured a comparable selection of model-agnostic XAI methods.
>
>
> ### Weakness 5
> No single method outperforms all others across all tested scores. However, the proposed ShapBPT method consistently ranks near the top compared to other methods. This also highlights that the various scores measure distinct aspects of model performance.
>
>
> *[Teneggi2022] Jacopo Teneggi, Alexandre Luster, and Jeremias Sulam. Fast hierarchical games for image explanations. IEEE Transactions on Pattern Analysis and Machine Intelligence, 45(4):4494–4503, 2022.*

---

### Official Review · Reviewer_ErUt · 2024-11-04

**Soundness:** 3
**Presentation:** 3
**Contribution:** 3
**Rating:** 6
**Confidence:** 5

**Summary:**

This paper introduces ShapBPT, a novel method for explainable AI designed to calculate an Owen approximation of Shapley coefficients. The approach employs a data-aware binary hierarchical structure, derived from a Binary Partition Tree (BPT), to improve the accuracy and efficiency of Shapley value computation. The authors validate the effectiveness of ShapBPT through extensive theoretical analysis and experimental results.

**Strengths:**

- The paper is clear and easy to follow
- The proposed method is novel and makes a meaningful contribution to the field of explainable AI by advancing Shapley-based explanation techniques.
- Theoretical insights are paired with empirical validation, demonstrating the efficacy of the approach across various settings.

**Weaknesses:**

- The method is only evaluated on pretrained classification models with different architectures, without comparisons to other training methods, such as supervised and self-supervised approaches.
- The experiments focus on image-based explanations, with no exploration of potential applications to video classification tasks.

**Questions:**

- How might the findings from 1 on the limitations of Shapley values affect the explainability claims made in this work?
- Did the authors consider a comparison with attention-based mechanisms that identify image regions contributing to classification decisions?

1. On the failings of Shapley values for explainability
https://www.sciencedirect.com/science/article/abs/pii/S0888613X23002438
2. Attention-based Interpretability with Concept Transformers https://research.ibm.com/publications/attention-based-interpretability-with-concept-transformers
3. Attention Flows are Shapley Value Explanations https://arxiv.org/pdf/2105.14652

---

> ### Author Response · Authors · 2024-11-18
>
> **W1**
> We opted to use pre-trained models since the primary focus of this paper is not on the training process. Additionally, using pre-trained models enhances the reproducibility of the evaluation.
>
> **W2**
> Video classification tasks are definitely an interesting application. However, implementing such tasks would likely require a form of time-coherent BPT hierarchy, which requires some advancements in the BPT algorithm family.
>
> ### Question 1
> Most of the claims in [1] are relevant to this work. The proposed method integrates data-aware hierarchies with the Owen formula, an approximation of the Shapley value formula. This approach is advantageous because it allows efficient computation of these approximations using data-aware hierarchies, enabling fast region identification. However, it is important to note that Owen scores remain approximations of the Shapley values.
>
> **Action**
> We will further clarify the limitations of approximate Shapley values in the final version. It is worth emphasizing, however, that this limitation is not unique to our method but is a general challenge for Shapley-based XAI methods.
>
> ### Question 2
> In this paper, we chose to focus on a model-agnostic setting to facilitate a uniform comparison of multiple XAI methods. Nevertheless, we included a comparison with a ViT model in Experiment E5, treating it as a black-box classifier.
>
> **Action**
> Although we are not thoroughly familiar with the Concept Transformer proposed in [2], we can explore the possibility of conducting an experimental comparison with ShapBPT. If the outputs of the Concept Transformer can be meaningfully compared with the saliency maps generated by ShapBPT, we will include this analysis in the final version of the paper.

---

### Meta-Review · Area_Chair_5mS6 · 2024-12-18

**Metareview:**

The paper has initially mixed reviews and no strong support, and the major concerns:

1. only compared on pre-trained classifiers, but do not compare different training methods [ErUt]
2. how to apply to video classification? [ErUt]
3. how do limitations of shapley [1] affect the claims of this work? [ErUt]
4. compare with attention-based mechanisms, only compared with simple segm methods [ErUt, EuK2]
5. limited number of images for evaluation [iXdf]
6. significance test is needed for Fig 5 [iXdf]
7. use of GT for evaluation in Figure 4 [ZEi3]
8. limitation of only one object [ZEi3]
9. why use BPT vs other segmentation methods? [ZEi3, EuK2]
10. limited novelty - using BPT instead of simpler segmentation [EuK2]
11. model is biased towards evaluations that are GT segment oriented (IoU), and evaluation is tuned to that  [EuK2]

The authors wrote a response, and after the discussion period, the paper still did not receive strong support and had mixed reviews. Reviewer ZEi3 still thought that there should be an evaluation on more complicated scenes (MSCOCO).

Of particular concern is Point 6 and the results in Figure 5, where the method only outperforms others on one perturbation metric (out of 4), and is eclipsed by other methods, including the SHAP partition explainer (AA, the baseline method). It does well on the metrics based on GT segments (max-IoU, AU-IoU), but as pointed out by EuK2 and ZEi3 (Points 11, 7), these are not particularly convincing metrics since 1) they ignore background as important regions; 2) it assumes that semantic segments are the important regions, but this is not always true; 3) it ignores higher-order interactions.

The AC agrees with these concerns, and thus recommends reject.

**Additional Comments On Reviewer Discussion:**

see above

---

### Decision · Program_Chairs · 2025-01-22

Reject